# UNDERSTANDING ATTENTION MECHANISMS

## ABSTRACT

Attention mechanisms have advanced the state of the art in several machine learning tasks. Despite significant empirical gains, there is a lack of theoretical analyses on understanding their effectiveness. In this paper, we address this problem by studying the landscape of population and empirical loss functions of attention-based neural networks. Our results show that, under mild assumptions, every local minimum of a two-layer attention model has low prediction error, and attention models require lower sample complexity than models not employing attention. Additionally for the popular self-attention, our theoretical results provide several guidelines for designing attention mechanisms. Our findings are validated with satisfactory experimental results on MNIST and IMDB reviews dataset.

## 1 INTRODUCTION

Significant research in machine learning has focused on designing network architectures for superior performance, faster convergence and better generalization. Attention mechanisms are one such design choice that is widely used in many natural language processing and computer vision tasks. Inspired by human cognition, attention mechanisms advocate focusing on the relevant regions of input data to solve a desired task rather than ingesting the entire input.

Several variants of attention mechanisms have been proposed, and they have advanced the state of the art in machine translation (Bahdanau et al., 2014; Luong et al., 2015; Vaswani et al., 2017), image captioning (Xu et al., 2015), video captioning (Pu et al., 2018), visual question answering (Lu et al., 2016), generative modeling (Zhang et al., 2018), etc. In computer vision, spatial/ spatio-temporal attention masks are employed to focus only on the relevant regions of images/ video frames for the underlying downstream task (Mnih et al., 2014). In natural language tasks, where input-output pairs are sequential data, attention mechanisms focus on the most relevant elements in the input sequence to predict each symbol of the output sequence. Hidden state representations of a recurrent neural network are typically used to compute these attention masks. The most popular implementation of this paradigm is self-attention (Vaswani et al., 2017), which uses correlation among the elements of the input sequence to learn an attention mask.

Substantial empirical evidence demonstrating the effectiveness of attention mechanisms motivates us to study the problem from a theoretical lens. In this work, we attempt to understand the loss landscape of neural networks employing attention. Analyzing the loss landscape and optimization of neural networks is an open area of research, and is a challenging problem even for two-layer neural networks (Poggio & Liao, 2017; Rister & Rubin, 2017; Soudry & Hoffer, 2018; Zhou & Feng, 2017; Mei et al., 2018b; Soltanolkotabi et al., 2017; Ge et al., 2017; Nguyen & Hein, 2017a; Arora et al., 2018). Convergence of gradient descent for two-layer neural networks has been studied in (Allen-Zhu et al., 2019; Mei et al., 2018b; Du et al., 2019). Ge et al. (2017) shows that there is no bad local minima for two-layer neural nets under a specific loss landscape design. Unfortunately, these results cannot directly be applied to attention mechanisms, as attention modifies the network structure and introduces additional parameters which are jointly optimized with the model. To the best of our knowledge, our work presents the first theoretical analysis on attention-based models.

Our main result shows that, under some mild conditions, every stationary point of attention models achieve a low prediction error. We perform an asymptotic analysis where we show that expected prediction on error goes to 0 as $n \to \infty$. We also show that attention models achieve lower sample complexity than the models not employing attention. We then discuss how the result can be extended to recurrent attention and multi layer cases, and discuss the effect of regularization. In addition, we

show how attention further helps improve the loss landscape by studying three properties: number of linear regions, flatness of local minima and small sample size training. We validate our theoretical results with experiments on MNIST and IMDB reviews dataset.

## 2 ATTENTION MODELS

Attention mechanisms are modules that help neural networks focus only on the relevant regions of input data to make predictions. To study such behavior, we analyze different types of attention models. We start with a naive global attention model. Then we analyze the most popular self-attention model and discuss the extension to recurrent attention in Appendix. In the naive global attention model, we consider a dataset $D = \{\boldsymbol{x}_i, y_i\}_{i=1}^N$, $\boldsymbol{x}_i \in \mathbb{R}^p$, $y_i \in \mathbb{R}$, where the output $y_i$ depends only on certain regions of input $\boldsymbol{x}_i$, i.e., $y_i = f^\star(\boldsymbol{a}^\star \odot \boldsymbol{x}_i)$, where $\boldsymbol{a}^\star$ is an attention mask, and $f^\star(.)$ is the ground-truth function that is used to generate the dataset and the vector $\boldsymbol{a}^\star \in [0,1]^p$. The set of entries $\{a_i^\star | a_i^\star \neq 0\}$ corresponds to the relevant region of the input, while the complementary set $\{a_i^\star | a_i^\star = 0\}$ corresponds to the irrelevant region.

We consider a two-layer ReLU-based neural network to approximate the function $f^*$. The network architecture consists of a linear layer followed by rectified linear units (ReLU) as a non-linearity and a second linear layer. Denote the weights of the first layer by $\boldsymbol{w}^{(1)}$, the weights of the second layer by $\boldsymbol{w}^{(2)}$, and the ReLU function by $\phi(\cdot)$. Then the response function for the input $\boldsymbol{x}$ can be written as $f(\boldsymbol{x}) = \boldsymbol{w}^{(2)T}\phi(\langle \boldsymbol{w}^{(1)}, \boldsymbol{x} \rangle)$. We call the above function "baseline model" since it does not employ any attention. To incorporate attention, we introduce the attention mask $\boldsymbol{a}$ as additional neural network parameters. The attention model we use can be written as:

$$f(\boldsymbol{x}) = \boldsymbol{w}^{(2)T}\phi(\langle \boldsymbol{w}^{(1)}, \boldsymbol{x} \odot \boldsymbol{a} \rangle) \tag{1}$$

In this paper, we focus on the regression task which minimizes the following loss function:$L = \mathbb{E}_{(\boldsymbol{x},y)\sim D}\|f(\boldsymbol{x}) - y\|_2^2$. While we present analysis on the regression task, our theory can easily be extended to classification tasks as well.

After a thorough analysis of global attention, we analyze a more practical self-attention setup, which comes from the transformer model proposed in Vaswani et al. (2017). The input $\boldsymbol{x}_i = (\boldsymbol{x}_i^1, \ldots, \boldsymbol{x}_i^p) \in \mathbb{R}^{t \times p}$, where $\boldsymbol{x}_i^j$ are t-dimensional vectors. Each $\boldsymbol{x}_i$ corresponds to independent sentences for $i = 1, \ldots, n$, and $\boldsymbol{x}^j$'s are the fixed dimensional vector embedding of each word in sentence $\boldsymbol{x}$. $\boldsymbol{w}^Q, \boldsymbol{w}^K \in \mathbb{R}^{d_q \times t}$ are the query and key matrices, and $\boldsymbol{w}^V \in \mathbb{R}^{d_v \times t}$ is the value matrix. For each input $\boldsymbol{x}_i$, the key is calculated as: $\boldsymbol{K}_i = (\boldsymbol{w}^K \boldsymbol{x}_i)^T \in \mathbb{R}^{p \times d_q}$; For $z^{th}$ vector in the input, the query vector is computed as: $\boldsymbol{Q}_i^z = (\boldsymbol{w}^Q \boldsymbol{x}_i^z)^T \in \mathbb{R}^{1 \times d_q}$ for $z = 1, \ldots, p$. The value matrix $\boldsymbol{V} = \boldsymbol{w}^V \boldsymbol{x}_i \in \mathbb{R}^{d_v \times p}$. Then the self-attention w.r.t to the $z^{th}$ vector in the input $\boldsymbol{x}_i$ is computed as:

$$\boldsymbol{a}_i^{self(z)}(\boldsymbol{x}_i^z, \boldsymbol{w}^Q, \boldsymbol{w}^K) = softmax(\frac{\boldsymbol{Q}_i \boldsymbol{K}_i^T}{\sqrt{d_k}}) \tag{2}$$

for $z = 1, \ldots, p$. And $\boldsymbol{a}_i^{self} = (\boldsymbol{a}_i^{self(1)}, \ldots, \boldsymbol{a}_i^{self(p)})$. This self-attention vector represents the interaction between different words in each sentence. The value vector for each word in the sentence $\boldsymbol{x}_i^z$ can be calculated as $\boldsymbol{V}_i^z = \boldsymbol{V}\boldsymbol{a}_i^{self(z)} \in \mathbb{R}^{d_v}$. This value vector is then passed to a 2-layer MLP parameterized by $\boldsymbol{w}^{(1)} \in \mathbb{R}^{pd_v \times d}$ and $\boldsymbol{w}^{(2)} \in \mathbb{R}^{d \times 1}$, resulting in the following generative model:

$$y_i = \boldsymbol{w}^{(2)\star T}\phi(\langle \boldsymbol{w}^{(1)\star}, vec(\boldsymbol{w}^V \boldsymbol{x}_i \boldsymbol{a}_i^{self}) \rangle) + \epsilon_i \tag{3}$$

where $vec(\cdot)$ represents the vectorization of a matrix. We also discuss the extension to recurrent attention model and multi-layer self-attention model in Appendix due to the page limit.

Note that naive global attention is not widely used in practice, because here we assume the attention mask is globally fixed for all points, but not a function of input. In real world application, the attention weights depend on the input, such as the self-attention framework we just introduced. Despite the limitation of naive global attention model, it is a fundamental building block of attention models, and needed to be analyzed first for two following reasons.

First, this fixed attention shares the core idea of attention: There is a specific intrinsic structure in data, and this intrinsic structure requires that we should assign different weights to different input features accordingly. And this weight assigning strategy should be learned from data. In naive

global attention, attention weights $\boldsymbol{a}$ are parameters themselves; In the self-attention model, the attention weights $\boldsymbol{a}$ depend on a function parameterized by value/key/query matrices, and we learn these matrices as attention parameters. And for both global and self-attention, we jointly learn the network weights($\boldsymbol{w}^{(1)}, \boldsymbol{w}^{(2)}$) and attention parameters($\boldsymbol{a}$ in global attention, value/key/query matrices in self-attention) at the same time. Therefore this naive global attention is a good starting point for analyzing attention mechanisms.

Second, we can gain helpful insights on attention by analyzing the global attention case. The number of non-zero elements of $\boldsymbol{a}$ in global attention represents both the size of attention parameters and the sparsity level of attention. In the standard self-attention model, size of attention parameters is determined by the size of value/key/query matrices. And the sparsity level is how many words we allow one word to attend to. By studying the effect of this quantity, we can have a better understanding of how the sparsity and parameter size of attention affect the model performance and sample complexity. The detailed discussions can be found in Section 3.

## 3    Loss landscape analyses on attention models

In Section 3.1, we analyze the loss landscape for the the naive global attention model, which is defined as

$$\min_{\boldsymbol{w}^{(1)}, \boldsymbol{w}^{(2)}, \boldsymbol{a} \in \mathcal{S}} \frac{1}{2n} \sum_{i=1}^{n} (\boldsymbol{w}^{(2)T} \phi(\langle \boldsymbol{w}^{(1)}, \boldsymbol{x}_i \odot \boldsymbol{a} \rangle) - y_i)^2 \tag{4}$$

where $\boldsymbol{w}^{(1)} \in \mathbb{R}^{p*d}$, $\boldsymbol{w}^{(2)} \in \mathbb{R}^d$, $\boldsymbol{a} \in \mathbb{R}^p$, and $\mathcal{S}$ is the parameter space of $\boldsymbol{a}$. Section 3.2 extends the loss landscape analysis to the self-attention model with $\boldsymbol{a} = \boldsymbol{f}(\boldsymbol{x})$. Section 3.3 discusses the extension to recurrent attention model. Section 3.4 discusses the effect of regularization and how the result can be extended to multi-layer networks.

To approximate the non-differentiable ReLU $\phi$, we use the softplus activation function $\phi_\tau(x)$, i.e.,

$$\phi_\tau(x) = \frac{1}{\tau} \log(1 + e^{\tau x})$$

Note $\phi_\tau$ converges to ReLU as $\tau \to \infty$ (Glorot et al., 2011). Theoretical results with $\phi_\tau(x)$ will hold for any arbitrarily large $\tau$. For ease of notation, we still use $\phi(x)$ to denote the softplus function.

### 3.1    Asymptotic property of naive global attention model

We first analyze the asymptotic prediction error of local minimum with large sample size. Let the covariance matrix of $\phi(\boldsymbol{w}^{(1)}, \boldsymbol{x} \odot \boldsymbol{a})$ be $(\boldsymbol{\Sigma}_\phi)_{ij} = cov(\phi_i(\langle \boldsymbol{w}^{(1)}, \boldsymbol{x} \odot \boldsymbol{a} \rangle), \phi_j(\langle \boldsymbol{w}^{(1)}, \boldsymbol{x} \odot \boldsymbol{a} \rangle))$ and $h(x_{\cdot k}) = x_{\cdot k} a_k (\boldsymbol{w}^{(2)} \odot \phi'(\langle \boldsymbol{w}^{(1)T}, \boldsymbol{x} \odot \boldsymbol{a} \rangle))$, where $x_{\cdot k}$ represents $k^{th}$ feature in $\boldsymbol{x}$, and $a_k$ is the attention mask for $x_{\cdot k}$. $\phi'(\langle \boldsymbol{w}^{(1)T}, \boldsymbol{x} \odot \boldsymbol{a} \rangle)$ is the first order derivative with respect to the value in $\phi(\cdot)$ and it belongs to $\mathbb{R}^d$. Let $u = (\boldsymbol{w}^{(2)} \phi(\langle \boldsymbol{w}^{(1)}, \boldsymbol{x} \odot \boldsymbol{a} \rangle) - \mathbb{E}(y|\boldsymbol{x}))$. Before proceeding, we introduce several necessary assumptions:

(A1) $\boldsymbol{x}_i$ are i.i.d with $\|\boldsymbol{x}_i\|_\infty < C_x$ for $i = 1, 2, ..., n$.

(A2) There exist $C_1, C_2$ such that $\|\boldsymbol{w}^{(1)}\|_F < C_1$ and $\|\boldsymbol{w}^{(2)}\|_2 < C_2$ for any $\boldsymbol{w}^{(1)}, \boldsymbol{w}^{(2)}$ in $\mathcal{S}$.

(A3) The output $y$ can be specified by the two-layer neural network up to an independent sub-Gaussian error with variance $\sigma^2$, i.e., there exists a set of parameter $(\boldsymbol{a}^\star, \boldsymbol{w}^{(1)\star}, \boldsymbol{w}^{(2)\star})$, such that $y_i = \boldsymbol{w}^{(2)\star T} \phi(\langle \boldsymbol{w}^{(1)\star}, \boldsymbol{x}_i \odot \boldsymbol{a}^\star \rangle) + \epsilon_i$, where $\epsilon_i \sim subG(0, C_3^2)$ for i=1,2,...n, with $\boldsymbol{x}_i \perp\!\!\!\perp \epsilon_i$. And $\|\mathbb{E}(\phi(\langle \boldsymbol{w}^{(1)\star}, \boldsymbol{x}_i \odot \boldsymbol{a}^\star \rangle))\|_2 \leq C_4$

(A4) $\|\boldsymbol{a}\|_0 \leq s_0$ such that $s_0 \leq p$, which represents the sparsity of the attention model, and $0 \leq \boldsymbol{a}_i \leq 1$ for any $i = 1, \cdots, p$.

(A5) $\lambda_{min}(\boldsymbol{\Sigma}_\phi) \geq C_\phi$, and when the estimation of $\phi$ is inaccurate, i.e, $\mathbb{E}(\|\phi(\langle \boldsymbol{w}^{(1)}, \boldsymbol{x} \odot \boldsymbol{a} \rangle - \phi^\star(\langle \boldsymbol{w}^{(1)}, \boldsymbol{x} \odot \boldsymbol{a} \rangle)\|_2) \geq O(\gamma)$, then there exists a feature $x_{\cdot k}$ and for the $t^{th}$ element of $h_t(x_{\cdot k})$, it satisfies $sd(h_t(x_{\cdot k})) = O(1)$ and $cor(u, h_t(x_{\cdot k})) = O(1)$ for $\boldsymbol{w}^{(2)}$ in the form of $\boldsymbol{\Sigma}_\phi^{-1} \boldsymbol{r} + o(\gamma)$.

(A6) $\mathbb{E}(\boldsymbol{w}^{(2)} \phi(\langle \boldsymbol{w}^{(1)}, \boldsymbol{x} \odot \boldsymbol{a} \rangle)) - \mathbb{E}(y) = o(\gamma/\sqrt{s_0})$.

(A7) The sum of the weights $\|\boldsymbol{a}\|_1 = 1$.

These assumptions are mild when the dimension of $\boldsymbol{x}$ is large. The justification of these assumptions are provided in the appendices. Given these assumptions, we show in Theorem 1 that the sample complexity required to converge to a good minimum is reduced with attention mechanism.

**Theorem 1.** *Under (A1) to (A6), for any $\gamma > 0$, suppose*

$$n \gtrsim \frac{s_0^2 C_1^2 C_x^2 \eta^2}{\gamma^2} \log(\frac{s_0 \eta}{\gamma})(pd + p + d)$$

*where $\eta = C_1 C_2 C_x$. Then with probability tending to 1, any stationary point $(\tilde{\boldsymbol{a}}, \tilde{\boldsymbol{w}}^{(1)}, \tilde{\boldsymbol{w}}^{(2)})$ of the objective function (4) satisfies the following prediction error bound:*

$$\mathbb{E}(\tilde{\boldsymbol{w}}^{(2)T} \phi(\langle \tilde{\boldsymbol{w}}^{(1)}, \boldsymbol{x} \odot \tilde{\boldsymbol{a}} \rangle) - \mathbb{E}(y|\boldsymbol{x}))^2 \lesssim \gamma^2$$

**Remark:** The sample complexity bound of Theorem 3 provides helpful insight for understanding attention mechanisms. With the sparsity structure of attention mask $\boldsymbol{a}$, attention mechanisms constrain the parameters in a smaller space, thus reducing the variance and the covering number. This leads to lower sample complexity compared to the baseline model not employing attention. It is straightforward to calculate the sample complexity bound for the baseline model. To achieve the same error bound, we substitute $s_0$ with $p$ in the bound, and this results in a much larger value. When $s_0$ is fixed, we can see up to a log term, prediction error $\gamma$ is proportion to $n^{-1/2}$, which is the optimal rate of convergence in regression. This shows that the bound is tight in this aspect.

Next, we extend Theorem 1 to the attention model with an additional sum-to-one constraint (A7). The discussion of the following Corollary 1 is provided in the appendix.

**Corollary 1.** *Under (A1) to (A7), for any $\gamma > 0$, suppose $n \gtrsim \frac{C_1^2 C_x^2 \eta^2}{\gamma^2} \log(\frac{\eta}{\gamma})(pd + p + d)$, where $\eta = C_1 C_2 C_x$. Then with probability tending to 1, any stationary point $(\tilde{\boldsymbol{a}}, \tilde{\boldsymbol{w}}^{(1)}, \tilde{\boldsymbol{w}}^{(2)})$ of the objective function (4) satisfies the following prediction error bound: $\mathbb{E}(\tilde{\boldsymbol{w}}^{(2)T} \phi(\langle \tilde{\boldsymbol{w}}^{(1)}, \boldsymbol{x} \odot \tilde{\boldsymbol{a}} \rangle) - \mathbb{E}(y|\boldsymbol{x}))^2 \lesssim \gamma^2$.*

### 3.2 Asymptotic property of self-attention model

In this section, we extend our previous analysis to the self-attention model. In self-attention, the attention mask is no more fixed globally, but instead a function of the input. We begin by analyzing a self-attention model with a known attention function $\boldsymbol{f}(\cdot)$, in which the weight $\boldsymbol{a}$ is not optimized together with $\boldsymbol{w}$. Similar bound can be derived for the following model:

$$\min_{\boldsymbol{w}^{(1)}, \boldsymbol{w}^{(2)} \in \mathcal{S}} \frac{1}{2n} \sum_{i=1}^{n} (\boldsymbol{w}^{(2)T} \phi(\langle \boldsymbol{w}^{(1)}, \boldsymbol{x}_i \odot \boldsymbol{f}(\boldsymbol{x}) \rangle) - y_i)^2 \tag{5}$$

**Proposition 1.** *Under (A1) to (A6), suppose that $\boldsymbol{a} = \boldsymbol{f}(\boldsymbol{x})$. For any $\gamma > 0$, given the sample size $n \gtrsim \frac{s_0^2 C_1^2 C_x^2 \eta^2}{\gamma^2} \log(\frac{s_0 \eta}{\gamma})(pd + d)$, where $\eta = C_1 C_2 C_x$, with probability converging to 1, any stationary point $(\tilde{\boldsymbol{w}}^{(1)}, \tilde{\boldsymbol{w}}^{(2)})$ of the objective function equation 5 satisfies that: $\mathbb{E}(\tilde{\boldsymbol{w}}^{(2)T} \phi(\langle \tilde{\boldsymbol{w}}^{(1)}, \boldsymbol{x} \odot \tilde{\boldsymbol{a}} \rangle) - \mathbb{E}(y|\boldsymbol{x}))^2 \lesssim \gamma^2$.*

Proposition 1 implies that if the self-attention mask can be precisely computed, global attention results can be extended to self-attention ones. However, the function $\boldsymbol{f}(\cdot)$ is not necessary known, and needs to be learnt in real world applications. Therefore the self-attention setup as we introduced in section 2 is more desired in real-world setting. Denoting $\boldsymbol{w} = (\boldsymbol{w}^{(1)}, \boldsymbol{w}^{(2)}, \boldsymbol{w}^Q, \boldsymbol{w}^K, \boldsymbol{w}^V)$, the two-layer self-attention model can be estimated by:

$$\min_{\boldsymbol{w}} \frac{1}{2n} \sum_{i=1}^{n} (\boldsymbol{w}^{(2)T} \phi(\langle \boldsymbol{w}^{(1)}, vec(\boldsymbol{w}^V \boldsymbol{x}_i \boldsymbol{a}_i^{self}) \rangle) - y_i)^2 \tag{6}$$

We now introduce necessary assumptions for analyzing self-attention model.

(A8) There exist $C_5, C_6$ and $C_7$ such that $\|\boldsymbol{w}^Q\|_F \leq C_5$, $\|\boldsymbol{w}^K\|_F \leq C_6$, $\|\boldsymbol{w}^V\|_F \leq C_7$.

(A9) The output $y$ can be predicted by the two-layer network (3) with an independent sub-Gaussian error with variance $\sigma^2$, i.e, there exists a set of parameters $(\boldsymbol{a}^\star, \boldsymbol{w}^{(1)\star}, \boldsymbol{w}^{(2)\star})$ such that $y_i = \boldsymbol{w}^{(2)\star T}\phi(\langle \boldsymbol{w}^{(1)\star}, vec(\boldsymbol{w}^V \boldsymbol{x}_i \boldsymbol{a}_i^{self(z)})\rangle) + \epsilon_i$, where $\boldsymbol{a}^{self}$ is calculated by (2); $\epsilon_i \sim subG(0, C_4^2)$ for $i = 1, 2, ...n$, with $\boldsymbol{x}_i \perp\!\!\!\perp \epsilon_i$.

(A10) We assume (A5) and (A6) holds, substituting $\boldsymbol{x}_i \odot \boldsymbol{a}$ with $vec(\boldsymbol{w}^V \boldsymbol{x}_i \boldsymbol{a}_i^{self})$, and $h_t(x._k) = (vec(\boldsymbol{w}^V \boldsymbol{x}_i \boldsymbol{a}_i^{self}))._k (\boldsymbol{w}^{(2)T}) \odot \phi^{'}(\boldsymbol{w}^{(1)}, \boldsymbol{x}_i \odot \boldsymbol{a}))$, where $(vec(\boldsymbol{w}^V \boldsymbol{x}_i \boldsymbol{a}_i^{self}))._k$ is the $k$-th element of the value matrices.

The assumption (A9) states that self-attention model can correctly predict the conditional mean $E(y_i|\boldsymbol{x}_i)$. Note that (A9) encompasses a more expressive class of models than (A3), which includes the models used in practice such as the transformers. (A10) is parallel to (A5) and (A6). Under these assumptions, we can obtain its sample complexity as given by following theorem:

**Theorem 2.** *Under (A1), (A2), and (A8) to (A10), for any $\gamma > 0$, given the sample size:*

$$n \gtrsim \frac{\eta^2 C_1^2 C_x^2}{\gamma^2} \log(\frac{C_5 C_6 C_7 \eta}{\gamma})(pd_v d + d + 2d_q t + d_v t)$$

*where $\eta = C_1 C_2 C_x$, with probability tending to 1, any stationary point $(\tilde{\boldsymbol{w}}^{(1)}, \tilde{\boldsymbol{w}}^{(2)}, \tilde{\boldsymbol{w}}^Q, \tilde{\boldsymbol{w}}^K)$ of the objective function (6) satisfies that: $\mathbb{E}(\tilde{\boldsymbol{w}}^{(2)T}\phi(\langle \tilde{\boldsymbol{w}}^{(1)}, vec(\boldsymbol{w}^V \boldsymbol{x}_i \boldsymbol{a}_i^{self})\rangle)) - \mathbb{E}(y|\boldsymbol{x}))^2 \lesssim \gamma^2$*

**Remark:** Theorem 2 shows that with the help of self-attention, we can achieve consistent prediction under more expressive class of models (assumption (A9)) which considers the interactions between vectors in data. It is worth pointing out that both global attention model and baseline model do not have consistency for the class of models beyond the ones stated in (A3). In other words, consistent prediction on the data distribution generated from equation 3 using baseline and global attention models requires introducing larger parameter space, for example, using more layers of network or more units in each layer. Self-attention model, on the other hand, achieves the more accurate estimation by constraining the parameter space and input space. And parallel to the sparsity level $s_0$ in Theorem 1, a proper choice of value/query/key matrices can help reduce sample complexity. If a sparse attention(i.e. one word should attend to all words, but only some relevant words), the sample complexity can also be further reduced similar with Theorem 1.

### 3.3 EXTENSION TO RECURRENT ATTENTION NETWORK

Sample complexity analysis can be extended to recurrent neural networks. This is included in the appendix. The key messages from our analysis include: (1) A good design of recurrent framework can help the network converge to a good stationary point with small sample complexity, and (2) An arbitrarily complex framework increases the sample complexity. Since in real world, the optimal recurrent framework is unknown, careful design choice has to be made for obtaining good sample complexity.

### 3.4 DISCUSSION: REGULARIZATION AND BEYOND 2 LAYERS

So far our analyses provide some theoretical justification on how attention mechanisms help learn superior models. Furthermore, our analysis also suggests proper regularization is helpful in training an attention model. An $\ell_1$ regularization on attention weights and $\ell_2$ regularization on network weights are effective in reducing the sample complexity. We also find that imposing constraints and regularization on network weights can help remove sharp minima, and keep flat minima with good generalization. Detailed discussions on regularization are provided in Appendix Section D.1.

Also, Theorem 1 and 2 can be extended to multi-layer attention network, under the assumptions parallel to (A8), (A9) and (A10): There exists a correct multi-layer self-attention network can specify the model, and the bias and gradient with respect to network weights are not uncorrelated. Under these assumptions, we provide sample complexity bound for multi-layer self-attention models. And all the discussions and insights are applied to multi-layer models. Explicit assumptions, discussions and theorems are provided in Appendix Section D.2. We avoid multi-layer setting in main context because it leads to over-complicated derivations and assumption justifications, and will distract readers from main idea of the paper. We believe two-layer models are representative enough to provide theoretical evidence on why attention reduces sample complexity.

# 4 ON IMPROVING THE LANDSCAPE STRUCTURE

In this section, we further investigate three additional properties on how attention mechanisms improve the landscape of neural networks. First, we show that in global attention model, attention mechanisms reduce unnecessary number of linear regions and maintain a low approximation error; Second, we show that flatness properties of minima are retained when attention mechanisms are used for both global attention and self-attention. Furthermore, our analysis indicates that for attention models, smaller sample size suffices to converge to good minima which generalize well in prediction. Finally, we show that the perfect in-sample prediction on small sample size is also achieved in attention networks for both global attention and self-attention.

## 4.1 ON THE NUMBER OF LINEAR REGIONS

We first study how attention mechanisms affect the number of linear regions (Montufar et al., 2014) in a wide two-layer network, where the number of units in the hidden layer is larger than the sparsity of the attention mask matrix.

**Theorem 3.** *Assume $\|\boldsymbol{a}\|_0 = s_0$, which is the sparsity of the mask matrix, and the number of units in the hidden layer $n_1 > s_0$. Then the maximal number of linear regions of the function by a two-layer fully connected neural network with ReLU activation function, is lower bounded by $\lfloor \frac{n_1}{s_0} \rfloor^{s_0}$.*

**Remark:** The theorem implies that when appropriate attention mechanism is used, the number of linear regions reduces leading to a simpler landscape, yet the approximation error remains small. This leads to lower sample complexity for achieving a desired prediction error. More detailed discussion can be found in the appendices. The result of Theorem 3 also applies to the self-attention with different attention sparsity(i.e. allowing how many words we allow one word to attend to).

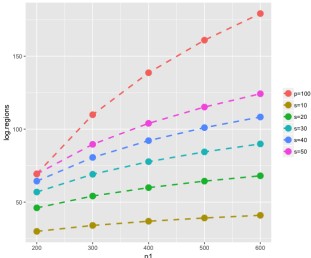

Figure 1: Number of linear regions in log scale v.s. sparsity

## 4.2 ON FLATNESS/SHARPNESS OF MINIMA

Many recent works, such as Keskar et al. (2016), argue that flatter local minima tend to generalize well. However, in a recent study, Dinh et al. (2017) observes that by scale transformation, the minima which are observationally equivalent, can be arbitrarily sharp, and the operator norm of a Hessian matrix can also be arbitrarily large. We will show that this fact also holds for the global attention mechanism, if no constraint on parameter $(\tilde{\boldsymbol{a}}, \tilde{\boldsymbol{w}}^{(1)}, \tilde{\boldsymbol{w}}^{(2)})$ is imposed. Here we introduce the definition of $\epsilon$-flatness as in Hochreiter & Schmidhuber (1997).

**Definition 1.** *Given $\epsilon > 0$, a minimum $\boldsymbol{\theta}$, and loss $L$, $C(L, \boldsymbol{\theta}, \epsilon)$ is the largest connected set containing $\boldsymbol{\theta}$ such that $\forall \boldsymbol{\theta}' \in C(L, \boldsymbol{\theta}, \epsilon)$, $L(\boldsymbol{\theta}') \leq L(\boldsymbol{\theta}) + \epsilon$, and its volume is called the $\epsilon$-flatness.*

In the following Theorem, we analyze the flatness of stationary point for both naive global and self-attention model.

**Theorem 4.** *(a) Consider the two-layer ReLU neural network with naive global attention in Section 3.1:*

$$y_i = \boldsymbol{w}^{(2)\star T} \phi(\langle \boldsymbol{w}^{(1)\star}, \boldsymbol{x}_i \odot \boldsymbol{a}^\star \rangle)$$

*and a minimum $\boldsymbol{\theta} = (\tilde{\boldsymbol{a}}, \tilde{\boldsymbol{w}}^{(1)}, \tilde{\boldsymbol{w}}^{(2)})$ satisfying that $\tilde{\boldsymbol{a}} \neq 0$, $\tilde{\boldsymbol{w}}^{(1)} \neq 0$, $\tilde{\boldsymbol{w}}^{(2)} \neq 0$. For any $\epsilon > 0$, $C(L, \boldsymbol{\theta}, \epsilon)$ has an infinite volume, and for any $M > 0$, we can find a stationary point such that the largest eigenvalue of $\nabla^2 L(\boldsymbol{\theta})$ is larger than $M$;*

*(b) Consider the two-layer ReLU neural network with self-attention mechanism as stated in Section 3.2:*

$$y_i = \boldsymbol{w}^{(2)\star T} \phi(\langle \boldsymbol{w}^{(1)\star}, vec(\boldsymbol{w}^V \boldsymbol{x}_i \boldsymbol{a}_i^{self}) \rangle)$$

*and a minimum $\boldsymbol{\theta} = (\tilde{\boldsymbol{w}}^{(1)}, \tilde{\boldsymbol{w}}^{(2)}, \boldsymbol{w}^V, \boldsymbol{w}^Q, \boldsymbol{w}^K)$ satisfying that $\tilde{\boldsymbol{w}}^i \neq 0$ for $i = (1), (2), V, Q, K$. For any $\epsilon > 0$, $C(L, \boldsymbol{\theta}, \epsilon)$ has an infinite volume, and for any $M > 0$, we can find a stationary point such that the largest eigenvalue of $\nabla^2 L(\boldsymbol{\theta})$ is larger than M.*

Theorem 4 indicates that property on flatness of minima is maintained when attention mechanism is applied, and there exist good sharp minima, coinciding with the observation in Dinh et al. (2017). However, there is no guarantee that all sharp minima are good in generalization. Revisiting our analysis in Section 3, the restriction on the parameter space (see (A2)) help remove these sharp minima. Specifically, (A2) provides an upper bound on the magnitude of $(\boldsymbol{a}, \boldsymbol{w}^{(1)}, \boldsymbol{w}^{(2)})$ and (A5) bounds the magnitude of $\phi(\langle \boldsymbol{w}^{(1)}, \boldsymbol{x}_i \odot \boldsymbol{a} \rangle)$ from below. These constraints control the parameter space and remove all sharp minima generated in Theorem 4 in which $\alpha_1$ or $\alpha_2$ goes to infinity. The $\ell_2$ bounds in (A2) can be achieved through a proper $\ell_2$ regularization (See Section 3.4).

### 4.3    ON SMALL SAMPLE SIZE

We conclude by studying the local minima of wide neural networks in small sample regime. (Nguyen & Hein, 2017b) proved that a two-layer neural network model can always achieve perfect empirical estimation error when the same size is small. Here, we extend this result for global and self-attention model. The discussion of Theorem 5 is deferred to the appendices.

**Theorem 5.** *(a) For naive global attention model, if $rank(\phi(\langle \tilde{\boldsymbol{w}}^{(1)}, \boldsymbol{x}_i \odot \boldsymbol{a} \rangle)_{i=1,2,..n}) = n$. Then every stationary point $(\tilde{\boldsymbol{a}}, \tilde{\boldsymbol{w}}^{(1)}, \tilde{\boldsymbol{w}}^{(2)})$ of the objective function (4), is a global minimum;*

*(b) For self attention model, if $rank(\phi(\langle \tilde{\boldsymbol{w}}^{(1)}, vec(\boldsymbol{w}^V \boldsymbol{x}_i \boldsymbol{a}_i^{self}) \rangle))_{i=1,2,..n}) = n$. Then every stationary point $(\tilde{\boldsymbol{w}}^{(1)}, \tilde{\boldsymbol{w}}^{(2)}, \tilde{\boldsymbol{w}}^V, \tilde{\boldsymbol{w}}^Q, \tilde{\boldsymbol{w}}^K)$ of object function (6) is a global minimum.*

## 5    EXPERIMENTS

### 5.1    SAMPLE COMPLEXITY

#### 5.1.1    GLOBAL ATTENTION MODEL

Theorem 1 proves that attention models require a lower sample complexity than baseline models, i.e., attention models require fewer samples to achieve the same test error as baseline models. This result is validated empirically in this experiment. To mimic the assumptions of Theorem 1, we consider ground truth two-layer neural network $G^*(\boldsymbol{x})$ is formed using random weights as inputs. The network $G^*$ maps the input vector $\boldsymbol{x} \in \mathbf{R}^{256}$ to 10-dimensional output. The input vector $\boldsymbol{x}$ is randomly sampled such that each element is drawn i.i.d from $\mathcal{N}(0, 1)$. An attention mask $\boldsymbol{a}^*$ is then constructed with $k$ randomly chosen elements as $1$ and the rest as $0$. The ground-truth labels are generated from $\boldsymbol{y} = G^*(\boldsymbol{x} \odot \boldsymbol{a}^*)$.

Table 1: Experiments on sample complexity for global attention model. $n_p$ denotes the number of trainable parameters in each model. All results are averaged over 5 runs

| Number of training samples | Test loss: baseline ($n_p = 34186$) | Test loss: attention ($n_p = 34442$) | Test loss: regularized attention ($n_p = 34442$) |
| --- | --- | --- | --- |
| 10000 | 2.1063 | 0.5484 | 0.0143 |
| 14000 | 0.4109 | 0.0382 | 0.0107 |
| 16000 | 0.1811 | 0.0211 | 0.0100 |
| 18000 | 0.1072 | 0.0163 | 0.0122 |
| 20000 | 0.0769 | 0.0101 | 0.0098 |
| 50000 | 0.0511 | 0.0060 | 0.0072 |

To test the sample complexity, we generate multiple datasets, each containing 10k, 14k, 16k, 18k, 20k and 50k unique samples respectively using the scheme mentioned in the previous section. A common test set of 5000 samples is created to evaluate each of the models. A regression model is then trained on each of these datasets. All models are trained with SGD optimizer with a fixed learning rate of $10^{-3}$. Table 1 reports test errors for baseline and attention models at 400k iterations

as the number of training samples vary. We observe that attention models need fewer training samples than baseline models to achieve a desired error. For instance, to attain the desired error $0.07$, attention models need $14000$ samples, whereas baseline models need $20000$ samples. We would like to point out that improvements obtained by attention models is not because of increase in model parameters. As shown in Table 1, the number of parameters in baseline and attention models are comparable. Hence, the performance gain is solely due to the attention mechanism.

**Regularization:** In section 3.4, we discuss how regularizing attention vector helps obtain a better attention model. To empirically validate this claim, we train a model with $L_1$ regularization on the attention vector. Same experimental setting as Section 5.1 is used, a regression model is trained using a two-layer neural network. We add the $L_1$ penalty on the attention vector to the objective: $L_{reg} = \sum_i |a_i|$. The results are also shown in Table 1. We observe that models trained with $L_1$ regularization achieves better sample complexity than its unregularized counterpart. We also observe the faster convergence when the models were regularized, as shown in the appendices.

### 5.1.2 SELF-ATTENTION MODEL

We extend the sample complexity experiments to self-attention model discussed in Section 3.2. Since our model is tailored towards natural language tasks, we consider the problem of sentiment classification on IMDB reviews dataset (Maas et al., 2011). Note that our analysis needs fixed length sentences which hardly holds true in any NLP dataset. So, we zero-pad all our sentences to make their length equal the maximum sentence length in the dataset ($2142$ for IMDB reviews). For every input word, we first obtain their corresponding pre-trained GloVE embeddings ($\in \mathbb{R}^{100}$) which is then passed to the neural network. As a baseline model, we flatten the input to one large vector of dimension $2142 \times 100$ and pass it to a 1-hidden layer MLP with 256 hidden units. For self-attention model, we use $\boldsymbol{w}^Q \in \mathbb{R}^{100 \times 100}$, $\boldsymbol{w}^K \in \mathbb{R}^{100 \times 100}$, $\boldsymbol{w}^V \in \mathbb{R}^{100 \times 2142}$. Once the attended features are computed per equation 2, it is passed to a 1-hidden layer MLP with 256 hidden units as the baseline model. All models were trained using Adam optimizer with learning rate $10^{-3}$. This was the setting that gave the best performance among optimizer and learning rate configurations we tried. A comparison of sample complexity of baseline model and the self-attention model is provided in Table 2. We clearly observe that self-attention model requires low sample complexity to achieve the same error as the baseline model. To test if improvements are obtained in attention model due to increase in model parameters, we ran the baseline model with twice the number of parameters as the self-attention model. Even with a large parameter size, baseline model performs poorly compared to self-attention models.

Table 2: Experiments on sample complexity for self attention model. $n_p$ denotes the number of parameters used in each model. All results are averaged over 5 runs.

| Number of training samples | Test accuracy (in %) | | |
| --- | --- | --- | --- |
| | Baseline $(n_p = 57335913)$ | Baseline $(n_p = 111744625)$ | Self-attention $(n_p = 57365913)$ |
| 875 | 63.38 | 64.38 | 70.52 |
| 1750 | 64.48 | 65.21 | 83.10 |
| 3500 | 63.51 | 64.32 | 86.14 |
| 5250 | 71.92 | 69.51 | 87.96 |
| 7000 | 75.49 | 76.32 | 87.10 |
| 8750 | 78.64 | 78.66 | 88.58 |
| 13125 | 79.97 | 79.72 | 88.98 |
| 17500 | 80.85 | 80.52 | 88.59 |

### 5.2 CONVERGENCE PLOT

This experiment studies the convergence of the empirical risk of the baseline and attention models. A modified MNIST dataset called NoisyMNIST is constructed where the images of digits from the MNIST dataset is embedded in noise as shown in Panel (a) of Figure 2. We consider the classification task to predict the labels of the digit in each image. Since the ground truth label depends only on certain regions of input, NoisyMNIST mimics the data generating process we consider in this paper.

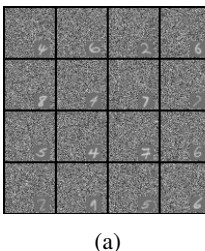
(a)

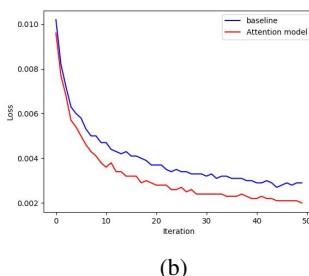
(b)

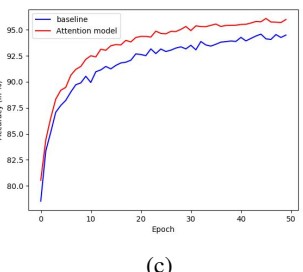
(c)

Figure 2: Visualization of NoisyMNIST in (a) and convergence plots for baseline and attention models on NoisyMNIST, including (b) plot of test loss and (c) plot of test accuracy over iterations.

For the baseline model, we train a two-layer neural network with 128 hidden units is using stochastic gradient descent. For attention models, the input tensor is multiplied element-wise with a learned attention mask $a$ as in equation 1. The attended input is then passed to the two-layer network. The convergence plots for baseline and attention models are plotted in Figure 2. We observe that the attention model converges faster, and to a better minimum than the baseline model. Similar behavior is observed for different learning rate and scale configurations as shown in the appendices.

## 5.3 HESSIAN PLOT

We study the Hessian matrix of the loss surface to validate the loss landscape of attention models. The same classification setup as the previous experiment is considered. The following two-layer neural network architecture was employed: $576 \rightarrow 16 \rightarrow 10$. The Hessian matrix of loss landscape about the computed minimum was, respectively, computed for the baseline and attention models, and their top $k$ sorted eigenvalues are plotted in Figure 3. Baseline models exhibit higher eigenvalues than attention models, so the loss landscape of attention models are flatter than the baseline models. Since flat landscapes lead to better generalization, models with attention generalize better than models without attention as shown in Section 5.2.

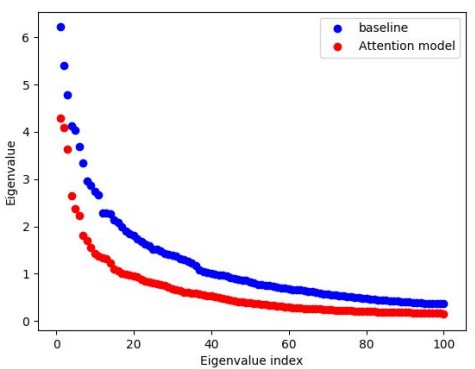

Figure 3: Top 100 eigenvalues of the Hessian matrix for baseline and attention models.

## 6 CONCLUSIONS

In this paper, we study the loss landscape of two-layer neural networks on global and self attention models, and show that attention mechanisms help reduce the sample complexity and achieve consistent predictions in the large sample regime. Additionally, by analyzing the number of linear regions, the loss landscape under small sample regime, and flatness of local minima, we demonstrate that attention mechanisms produce a well behaved loss landscape that leads to a good minima. Extensive empirical studies on NoisyMNIST dataset and IMDB reviews dataset validate our theoretical findings.

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

# Appendices: Understanding Attention Mechanisms

In appendices, Section A presents the extensions of our analyses to recurrent attention model; Section B provides detailed justification of our assumptions; Section C discusses the implications of theorem results in more detail(Theorem 1, Corollary 1, Theorem 3 and Theorem 5 in order); Section D discusses the regularization effect in attention networks and potential extensions beyond 2 layers and to CNN/RNN. Finally We provide proofs and additional experiment results in section E and F separately.

## A    EXTENSION TO RECURRENT ATTENTION MODEL

Here we consider analyzing the representative recurrent attention framework in Bahdanau et al. (2014). In the recurrent attention network, we still have each data point $\boldsymbol{x}_i = (\boldsymbol{x}_i^1, \ldots, \boldsymbol{x}_i^p) \in \mathbb{R}^{t \times p}$, corresponding to $p$ words with $t$-dimensional embedding. Then the generative model can be represented as:

$$y_i = \boldsymbol{w}^{(2)T} \langle \boldsymbol{w}^{(1)}, \sum_{j=1}^{t} \boldsymbol{a}(\boldsymbol{x}_i)_j \boldsymbol{x}_i^j \rangle + \epsilon_i$$

Analogous to NLP setting, $\boldsymbol{a}(\boldsymbol{x}_i)$ is a unknown function mapping $\boldsymbol{x}_i$ to a $t$-dimensional vector, where $\boldsymbol{a}(\boldsymbol{x}_i)_j$ represents the effect of the $j^{th}$ word in the sentence for point $i$. To simplify the model, we use data features themselves as their annotation, then for time stamp $k = 1, \ldots, T$, The recurrent attention model estimates $\boldsymbol{a}(\boldsymbol{x}_i)$ as follows:

$$s_k = \boldsymbol{f}(s_{k-1}, c_{k-1}); \ e_{kj} = score(s_{k-1}, \boldsymbol{x}_i^j)$$

$$\alpha_{kj} = \frac{e_{kj}}{\sum_{j=1}^{t} e_{kj}}; \ c_k = \sum_{j=1}^{t} \alpha_{kj} x_i^j$$

$$y_k = \boldsymbol{w}^{(2)T} \phi(\langle \boldsymbol{w}^{(1)}, c_k \rangle)$$

where $score(\cdot)$ is the scoring function representing how well the inputs around position $j$ and the output at position $i$ match. It can be dot product or MLP. And $\boldsymbol{f}(\cdot)$ is the function to update $s_k$. Suppose the parameter set inside these two functions are $\boldsymbol{w}_a$ and $\boldsymbol{w}_f$ with number of parameters as $d_a$ and $d_f$ accordingly. Here we show that when these two functions are expressive enough, recurrent attention network will also have sample complexity bound parallel to previous sections. Here we introduce necessary assumptions.

(A11) The output $y$ can be predicted by the two-layer network with an independent sub-Gaussian error with variance $\sigma^2$, i.e, there exists a set of parameters $(\boldsymbol{w}^{(1)\star}, \boldsymbol{w}^{(2)\star})$ such that $y_i = \boldsymbol{w}^{(2)\star T} \phi(\langle \boldsymbol{w}^{(1)\star}, \sum_{j=1}^{t} \boldsymbol{a}(\boldsymbol{x}_i)_j \boldsymbol{x}_i^j \rangle) + \epsilon_i$, where $\epsilon_i \sim subG(0, C_4^2)$ for $i = 1, 2, \ldots n$, with $\boldsymbol{x}_i \perp\!\!\!\perp \epsilon_i$.

(A12) Suppose (A5) holds when we substitute $\boldsymbol{x} \odot \boldsymbol{a}$ with $\sum_{j=1}^{t} \boldsymbol{a}(\boldsymbol{x}_i)_j \boldsymbol{x}_i^j$ and $h_t(x_{\cdot k}) = \boldsymbol{a}(\boldsymbol{x}_{ik}) \boldsymbol{x}_{ik}^j (\boldsymbol{w}^{(2)T}) \odot \phi^{'}(\langle \boldsymbol{w}^{(1)}, \boldsymbol{x}_i \odot \boldsymbol{a} \rangle))$.

(A13) We assume $\|\boldsymbol{w}_a\|_2 \le C_8$ and $\|\boldsymbol{w}_f\|_2 \le C_9$.

(A11) and (A13) are parallel to (A1) to (A5) in global attention case. They can be justified similar as them, which is discussed in Section B of appendices.

Now we can provide following sample complexity bound extended from previous sections.

**Theorem 6.** *Under (A1),(A2),(A11) to (A13), there exists a sufficient large T, for any $\gamma > 0$, suppose*

$$n \gtrsim \frac{C_1^2 C_x^2 \eta^2}{\gamma^2} \log(\frac{\eta C_8 C_9}{\gamma})(td + d + d_w + d_f)$$

*where $\eta = C_1 C_2 C_x$, such that if there exist stationary point(s), then with probability tending to 1, any stationary point $(\tilde{\boldsymbol{w}}^{(1)}, \tilde{\boldsymbol{w}}^{(2)}, \tilde{\boldsymbol{w}}_f, \tilde{\boldsymbol{w}}_a)$ satisfies the following prediction error bound:*

$$\mathbb{E}(\tilde{\boldsymbol{w}}^{(2)} \phi(\langle \tilde{\boldsymbol{w}}^{(1)}, \sum_{j=1}^{t} \boldsymbol{a}(\boldsymbol{x}_i)_j \boldsymbol{x}_i^j \rangle) - \mathbb{E}(y|\boldsymbol{x}))^2 \lesssim \gamma^2$$

**Remark:** This theorem provide a sample complexity bound for recurrent attention network. It holds when such 'good stationary point' exists. It also shows a trade-off between a complicated recurrent attention network and the sample complexity bound. If $\boldsymbol{f}(\cdot)$ and $\boldsymbol{a}(\cdot)$ are properly selected, they will be sufficient expressive to obtain good stationary points, and also the number of parameters $d_w$ and $d_f$ will not be too large. In this way. an ideal sample complexity bound to these good stationary points can be achieved as theorem says. However, with a over complicated design in these functions, the sample complexity bound will be large; With a over simple design, such good stationary points don't exist. It is parallel to a trade-off between approximation error and estimation error in learning theory. The theory implies a good design of recurrent structure will help achieve an optimal sample complexity in recurrent attention model.

## B   JUSTIFICATION OF ASSUMPTIONS (A1)–(A6)

In this section, we discuss the rationality of Assumptions (A1) to (A6). Note that (A1) to (A6) are required to prove the main result in Theorem 3 and they have also been studied by Keskar et al. (2016); Dinh et al. (2017); Mei et al. (2018a;b); Nguyen & Hein (2017b). In what follows, we will show in details that (A1)–(A6) are all reasonable assumptions.

First of all, (A1) and (A2) require upper bounds on the input $\boldsymbol{x}_i$ and $\ell_2$ bound for network weights. It is a standard assumption in landscape analysis (Mei et al., 2018a;b), and also it is crucial to remove sharp minima which may not generalize well (Keskar et al., 2016; Dinh et al., 2017).(See remark after Theorem 4). These assumptions can be achieved through regularization.

(A3) requires that this two-layer network are rich enough to specify the condition mean $E(y_i|\boldsymbol{x}_i)$. It has been studied that general bounded functions with a Fourier representation on $[-1, 1]$ can be well approximated by the defined two-layer network(Barron & Klusowski, 2018).

(A4) requires a sparse structure on $\boldsymbol{a}^\star$; otherwise, the model would be equivalent to the baseline model, simply just choose attention masks all being 1. And $\|\boldsymbol{a}\|_\infty \leq 1$ requires the attention weight ranges from 0 to 1.

(A5) is a technical condition for our analysis of stationary point. (A5) includes two parts. Both of them hold naturally when dimensionality is large. Firstly, the lower eigenvalue bound assumes $\phi(\langle \boldsymbol{w}^{(1)}, \boldsymbol{x}_i \odot \boldsymbol{a}\rangle)$ is not degenerated. If eigenvalue assumption is violated, the model is equivalent to a network with fewer number of units in hidden layers, and we can study this equivalent degenerated one instead; This assumption also guarantees us to remove sharp minima(Keskar et al., 2016; Dinh et al., 2017), same as (A1) and (A2). Secondly, (A5) assumes when $\phi(\langle \boldsymbol{w}^{(1)}, \boldsymbol{x}_i \odot \boldsymbol{a}\rangle)$ is not well estimated, there exists an 'active feature'. The correlation between this feature and bias cannot be cancelled by the direction of a specific linear combination of $\phi(\langle \boldsymbol{w}^{(1)}, \boldsymbol{x}_i \odot \boldsymbol{a}\rangle)$. Intuitively, this assumption says that the correlation between $h_t(x._k)$ and $E(y|x._k)$ cannot be fully explained by a fixed linear combination if there is some systematic bias in $\phi(\langle \boldsymbol{w}^{(1)}, \boldsymbol{x}_i \odot \boldsymbol{a}\rangle)$. Since there is systematic bias, it is reasonable to assume this systematic bias cannot be uncorrelated with all the directions spanned by $h_t(\boldsymbol{x})$. This correlation assumption between $h_t(\boldsymbol{x}._j)$ and $u$ is parallel to the full column rank condition in (Nguyen & Hein, 2017b). Considering all active terms in $h_t(x._k)$, they span a larger space comparing to $\phi(\langle \boldsymbol{w}^{(1)}, \boldsymbol{x}_i \odot \boldsymbol{a}\rangle)$, considering $d$ is a fixed dimension. Thus it is a natural assumption if we have reasonable large dimensionality and the model doesn't degenerate. What's more, with required large sample size, we can also straight forwardly evaluate this assumption by checking empirical correlation, and avoid this type of bad minima through random initialization and proper gradient descent type algorithm(Allen-Zhu et al., 2019). And it also will not affect the key structure of our proof.

Specifically, for the $sd(h_t(x._k)) = O(1)$ part, since $h_t(x._k) = x._k a_k(\boldsymbol{w}_t^{(2)} \cdot \phi'(\langle \boldsymbol{w}^{(1)}, \boldsymbol{x}_i \odot \boldsymbol{a}\rangle)_t)$, where $\phi'(\langle \boldsymbol{w}^{(1)}, \boldsymbol{x}_i \odot \boldsymbol{a}\rangle)_t$ are $O(1)$ with positive probability otherwise the network always predicts zero, therefore we have $sd(h_t(x._k)) = O(1)$ as long as $sd(x._k) = O(1)$ and $a_k = O(1)$. For the correlation assumption $cor(u, h_t(x._k)) = O(1)$, if this assumption doesn't hold, $\|cor(u, h(\boldsymbol{x}_{j_k}))\|_2 = o(1)$ for all $k = 1, \ldots, p$. By linear combination, for any vector $\boldsymbol{z} = (z_1, \ldots, z_k)$ with bounded $\ell_2$ norm, we have $cor(u, \boldsymbol{z}^T(\boldsymbol{x}_1 \odot \boldsymbol{a})((\boldsymbol{w}_t^{(2)} \odot \phi'(\langle \boldsymbol{w}^{(1)}, \boldsymbol{x} \odot \boldsymbol{a}\rangle)) = o(\gamma)$ for all $t = 1, \ldots, d$. We know for any $\bar{\boldsymbol{w}}^{(1)} \in \mathbb{R}^{p \times d}$, the term $\bar{\boldsymbol{w}}^{(1)}(\boldsymbol{x} \odot \boldsymbol{a})((\boldsymbol{w}_t^{(2)} \odot \phi'(\langle \boldsymbol{w}^{(1)}, \boldsymbol{x} \odot \boldsymbol{a}\rangle)$ can be represented as the combination of term $\boldsymbol{z}^T(\boldsymbol{x}_1 \odot \boldsymbol{a})((\boldsymbol{w}_t^{(2)} \odot \phi'(\langle \boldsymbol{w}^{(1)}, \boldsymbol{x} \odot \boldsymbol{a}\rangle))$. By the arbitrary choice of

our $z$, it means that all the directions of $z$ are almost not correlated with $u$. When $p$ and $s_0$ are high-dimensional, $\bar{w}^{(1)}(x \odot a)((w_t^{(2)} \odot \phi'(\langle w^{(1)}, x \odot a \rangle))$ spans all the directions as $p \to \infty$, thus there must be some direction is correlated with $u$. Therefore it is reasonable to assume a $O(1)$ correlation occurs.

(A6) assumes that we fit a model with expectation close to truth. We can always achieve this by centralization.

With these assumptions, we just make sure we remove unnecessary minima, such that we can concentrate on analyzing the behaviour of good stationary points of our interest. Assumptions have no influence on the main idea of our following theorem: showing the required sample complexity to approach a good minimum is reduced with attention mechanism.

**Remark:** The assumptions are also related to the question that when attention mechanism should be applied in network. (Zhou et al., 2015) shows that attention sometimes can be badly used in certain cases, which shares the similar philosophy with our analyses. For example, our theoretical analyses depend on the assumption (A3), that is, the model can be correctly specified when attention mechanism applies. This can be violated when all the variables are useful and they all need to be included in the model. In this case, the model with attention will be inconsistent. It indicates that in this case when assumptions are violated, neural networks can achieve the precise estimation only through the over-parameterization of $w^{(1)}$.

# C    DISCUSSIONS ON THEOREM RESULTS

## C.1    THEOREM 1: COMPARISON OF SAMPLE COMPLEXITY WITH BASELINE MODEL

When we compare sample complexity of attention model versus baseline model, we say the key difference will be that $s_0$ is substituted with $p$, which can be a larger diverging dimensionality. Constants $C_1$ and $C_2$ can be regarded remain same when we compare attention model with the baseline for following reasons: (1) We assume same generative model, and the network size is the same, thus the optimal weight is known to be same. To make sure trained network weight is on the same scale with the optimal one, it is fair to keep there $\ell_2$ bound constant same. (2) In this framework, we study the effect when $p$ diverges to $\infty$ as $n \to \infty$. In this aspect, we don't expect the weight norm also diverges, since a diverging weight leads to overfitting. By imposing $\ell_2$ regularizations on weights, we can always control the upper bound of $\ell_2$ norm. Therefore it is reasonable to assume its norm is bounded by a sufficient large constant. By imposing $\ell_2$ regularization on weights, we can always control the upper bound of $\ell_2$ norm by this sufficient large constant. (3) Even with overfitting, $C_2$ in baselines are expected to be even larger due to the overfitting effect(Ch.7.Goodfellow et al. (2016). It will further explain why sample complexity of baseline is even larger comparing with fixing $C_1$ and $C_2$. In the experiment result, we also observe that the $\ell_2$ norm of weights from baseline is larger or equal to the attention network.

## C.2    COROLLARY 1: SUM-TO-ONE GLOBAL ATTENTION

Given the assumption (A7), we do not need $s_0$ on the sample complexity bound for the sum-to-one global attention model. However, if we rescale $x$ properly, the result will be parallel to Theorem 1 result.

## C.3    THEOREM 3: NUMBER OF LINEAR REGIONS

In a sufficiently wide network, $\lfloor \frac{n_1}{s_0} \rfloor^{s_0}$ is much smaller than $\lfloor \frac{n_1}{p} \rfloor^{p}$. Then, $(\frac{n_1}{s_0})^{s_0} \leq (\frac{n_1}{p})^{p}$ holds as long as $n_1 \geq \exp(\frac{p \log p - s_0 \log s_0}{p - s_0})$. Given $\frac{p \log p - s_0 \log s_0}{p - s_0} \leq \frac{p}{p-s} \log p$, since $\frac{p}{p-s}$ is close to $p$ when $s$ is relatively small, the result still holds when $n_1$ is larger than the order of $p$.

For illustration, the bounds are plotted in Figure 1 in paper. The red line is for baseline model with $p = 100$, and others are attention model with different sparsity level $s_0$. In general we can see the bound for attention model is smaller than that for baseline model. The implication is that, when proper attention mechanism is applied, the approximation error remains small, and we can use a

simpler landscape structure with less number of linear regions to reduce the estimation error. This is the why we can achieve specific prediction error rate with a smaller sample complexity.

### C.4 THEOREM 5: SMALL SAMPLE SIZE RESULT

In assumptions, $rank(\phi(\langle \tilde{\boldsymbol{w}}^{(1)}, \boldsymbol{x}_i \odot \boldsymbol{a} \rangle)_{i=1,2,..n}) = n$ is a mild assumption in a wide network with over-parameterization. We can see that as long as we choose the number of units $d$ to be larger than $n$, the linear dependence of $\langle \boldsymbol{w}^{(1)}, \boldsymbol{x}_i \odot \boldsymbol{a} \rangle_{i=1,2,..n}$ holds with measure zero. In other words, almost surely this matrix has full column rank $n$. Thus after the nonlinear activation, The full column rank still holds almost surely. This assumption is similar to the condition in Theorem 3.8 of Nguyen & Hein (2017b), where the number of units in some layer is larger than the sample size. When the sample size is smaller than the number of units in the network, this theorem holds for the network without attention. It has been proved by Nguyen & Hein (2017b) and Soudry & Carmon (2016) under different conditions.

## D REGULARIZATION EFFECT AND MULTI-LAYER EXTENSIONS

Our analyses provide some theoretical justification on how attention mechanisms help learn superior models. The benefit mainly comes from that the attention weight $\boldsymbol{a}$ shrinks the whole parameter space, while this space is still large enough to capture all the necessary information. Thus the gradient and Hessian are more controllable in this space, and the landscape of loss function behaves better compared to the baseline model. It will be shown again in Theorem 4 in the following section. This fact holds for both global and self-attention model. Our theory further validates that as long as the attention masks are learnt well, the performance is expected to improve. And this effect can be more significant when the weight is sparse. We also find that imposing constraints and regularization on network weights can also help remove sharp minima, and keep flat minima with good prediction. These discussions can be found in D.1.

The main idea of Theorem 1 and 2 can also be extended to multi-layer networks under the assumption that bias and gradients of weights are not uncorrelated. They can be found in D.2.

### D.1 REGULARIZATION

Our analyses indicate that it is worth considering approaches to control the bounds for the network weight matrices and pursue more accurate estimation of attention weighted input space. Motivated by this message, we suggest two possible regularization methods, which help to improve currently used attention mechanisms.

- $\ell_2$ Regularization on $\boldsymbol{w}$

  It is known that in optimization, $\ell_2$ regularization on $\boldsymbol{w}$ is equivalent to specific $\ell_2$ bound for $\boldsymbol{w}$. For our two-layer model, if we choose regularization level properly, the $\ell_2$ bound for $\boldsymbol{w}^{(1)}$, $\boldsymbol{w}^{(2)}$( also $\boldsymbol{w}^Q$,$\boldsymbol{w}^K$ and $\boldsymbol{w}^V$ in self-attention model) will match our assumption. In practice, a proper regularization should be as large as possible, as long as a nice in-sample prediction is still achieved. This is important for attention mechanisms to keep prediction power while improving the landscape behavior of loss function.

- $\ell_1$ Regularization on $\boldsymbol{a}$ to achieve sparse attention weight

  As we discussed, $\ell_1$ regularization can help achieve sparsity and more precise estimation. For our 2-layer model, imposing $\ell_1$ regularization will not affect the analysis since we didn't use the gradient w.r.t $\boldsymbol{a}$ in the analysis. It means the same theoretical guarantee holds for the regularized network. At the same time $\boldsymbol{a}$ will be more sparse, practically lead to a more precise estimation, and more interpretable result.

  This idea can also be adapted to self-attention model. In self-attention, we can add a regularization to $\boldsymbol{a} = softmax(\frac{QK^T}{\sqrt{d_k}})$. With this regularization, only part of the value matrices proceeds to the decoding procedure. In real world application such as in transformer, the sparsity in $\boldsymbol{a}_i$ corresponds to the case when there are some words in a sentence should not attend each other. In this case, a sparse $\boldsymbol{a}^{self}$ help us only focus attention between useful words in a sentence, thus improve predition.

It is worth mentioning that, through our theorem, we can tell why $\ell_1$ loss are more helpful than $\ell_2$ loss under sparsity assumption. Both $\ell_1$ loss and $\ell_2$ loss will control the magnitude of attention mask, however, $\ell_1$ loss can also help control sparsity level $s_0$, and we can see that the sample complexity bound is proportion to $s_0^2$. Although $\ell_2$ loss on attention mask can also help reduce the sample complexity, its effect will only be reflected in the $\log$ term, therefore it is less effective. Similarly in Theorem 2, if we assume the attention to be sparse(i.e. one word should not attend to all words, but only some relevant words), then $\ell_1$ loss can help further reduce the sample complexity.

Regularization is one experimental approach for estimating a sparse and precise $\boldsymbol{a}$ and $\boldsymbol{w}$. If some other methods can achieve this target, they are also the right directions to reduce sample complexity and improve current attention structures.

## D.2   Beyond two layers

Here we discuss how our results of Theorem 1 and 2 can be extended to multi-layer neural nets.

First we consider a $D$-layer network with naive global attention structure: $f(\boldsymbol{x}) = (\boldsymbol{w}^{(D)}\phi(\boldsymbol{w}^{(D-1)}\ldots\phi(\langle\boldsymbol{w}^{(1)}, \boldsymbol{x}\odot\boldsymbol{a}\rangle))$. Denoting $\nabla f(\boldsymbol{w}^k) = \frac{\partial f(\boldsymbol{x})}{\partial\boldsymbol{w}^{(k)}}$, under the assumption that at least one gradient term $\nabla f(\boldsymbol{w}^k)$ for $k = 1, \ldots, D$ has correlation at rate $O(1)$ with bias $u$(this assumption is parallel to (A5)), we can show the expectation term $\mathbb{E}_{x,y}(\nabla R(\boldsymbol{w}^{(k)})) \geq O(\gamma)$. Then using the similar $\epsilon$-covering and uniform convergence technique as Theorem 1 goes, we can still show attention mechanism leads to a smaller $\epsilon$-covering number and tighter Hoeffding bound, thus leading to a smaller sample complexity bound comparing to baseline model.

Then we consider a $D$-layer network with self-attention structure. We denote the $k^{th}$ self-attention layer follows $g_k(\boldsymbol{x}_g^{k-1}) = \boldsymbol{w}^{k_2}\phi(\langle\boldsymbol{w}^{k_1}, \boldsymbol{w}^V\boldsymbol{x}_g^{k-1}\boldsymbol{a}^{self}\rangle))$, where $\boldsymbol{x}_g^{k-1}$ is the output of $(k-1)^{th}$ self-attention layer, with $\boldsymbol{w}^V \in \mathbb{R}^{d_v\times t}$, $\boldsymbol{x}_g^{k-1} \in \mathbb{R}^{t\times d_{k-1}}$, $\boldsymbol{a}^{self} \in \mathbb{R}^{d_{k-1}\times d_{k-1}}$, $\boldsymbol{w}^{k_1} \in \mathbb{R}^{d_v\times q_k}$ and $\boldsymbol{w}^{k_2} \in \mathbb{R}^{d_k\times q_k}$. $\boldsymbol{a}$ is calcualted in the same way with two-layer self-attention network. Then we have the final output $f(\boldsymbol{x}) = \boldsymbol{w}^{D_1}\phi(\langle\boldsymbol{w}^{D_1}, vec(\boldsymbol{w}^V\boldsymbol{x}_g^{D-1}\boldsymbol{a}^{self})\rangle)$, where $\boldsymbol{x}_g^{D-1} = (g_{D-1}(\cdots g_1(\boldsymbol{x}))$, and $\boldsymbol{w}^{D_1} \in \mathbb{R}^{\times}$, $\boldsymbol{w}^{D_2} \in \mathbb{R}^{\times}$. In this way, the network calculate self-attention $D$ times and finally produce the final prediction. It is worth mentioning that, To obtain a scalar prediction in regression model, we flatten the value matrix of the last layer as same as the two-layer model. We still denote $u = (\boldsymbol{w}^{D_2}\phi(\langle\boldsymbol{w}^{D_1}, vec(\boldsymbol{w}^V\boldsymbol{x}_g^{D-1}\boldsymbol{a}^{self})\rangle) - \mathbb{E}(y|\boldsymbol{x}))$. Then the necessary assumptions parallel to (A2), (A9) and (A10) are as follows.

- (A14) All weights $\boldsymbol{w}^{k_j}$ for $k = 1, \ldots, D$ and $j = 1, 2$ satisfy $\|\boldsymbol{w}^{k_j}\|_2 \leq C_{10}$. And we assume the prediction is centered, i.e. $E(u) = 0$.

- (A15) The output $y$ can be predicted by the $D$-layer self-attention network with an independent sub-Gaussian error with variance $\sigma^2$, i.e, there exists a set of parameters $(\boldsymbol{w}^{(1)\star}, \boldsymbol{w}^{(2)\star})$ such that $y_i = \boldsymbol{w}^{D_2}\phi(\langle\boldsymbol{w}^{D_1}, vec(\boldsymbol{w}^V\boldsymbol{x}_g^{D-1}\boldsymbol{a}^{self})\rangle) + \epsilon_i$ as defined, where $\epsilon_i \sim subG(0, C_4^2)$ for $i = 1, 2, ...n$, with $\boldsymbol{x}_i \perp\!\!\!\perp \epsilon_i$.

- (A16) There exists interger $k$ and $r$ such that $k \in \{1, \ldots, D\}$ and $r \in \{1, 2\}$, such that $cor(\nabla f(\boldsymbol{w}^{k_r}), \boldsymbol{u}) = O(1)$ and $sd(\nabla f(\boldsymbol{w}^{k_r})) = O(1)$.

Then we can have the following theorem of sample complexity bound of multi-layer self-attention model:

**Theorem 7.** *Under (A1) and (A14) to (A16), we assume the weight term $\boldsymbol{w}^{k_r}$ satisfies (A16) with $\|\nabla f(\boldsymbol{w}^{k_r})\|_2 \leq c_k$. $d_{self}$ is the total number of parameters in all value, query, key matrices. Then for any $\gamma > 0$, given the sample size:*

$$n \gtrsim \log(\frac{c_k}{\gamma})(d_{self} + \sum_{i=1}^{k}(d_k + d_v)q_k)$$

*where $\eta = C_1C_2C_x$, with probability tending to 1, any stationary point $(\tilde{\boldsymbol{w}}^{(1)}, \tilde{\boldsymbol{w}}^{(2)}, \tilde{\boldsymbol{w}}^Q, \tilde{\boldsymbol{w}}^K)$ of the objective function (6) satisfies that: $\mathbb{E}(f(\boldsymbol{x}) - \mathbb{E}(y|\boldsymbol{x}))^2 \lesssim \gamma^2$*

**Remark:** Because multi-layer self-attention models include a large parameter set with complicated gradients, the assumptions are not as intuitive as the two-layer model. But the main assumptions are parallel, such that the bias $u_i$ cannot be uncorrelated with all possible directions. And this assumption is reasonable considering the high-dimensionality nature of networks. The extension of this multi-layer model is omitted in the main paper since it leads to over-complicated derivations and complicated assumption discussion and will distract readers from main ideas of the paper. We believe the two-layer attention model is representative enough to provide theoretical evidence on why attention reduces sample complexity.

### D.3 CNN AND RNN

Our analyses are based on fully connected network, and may not be able to apply to more involved task such as CNN and RNN directly. But we believe the key message of our analyses also provide insights in analyzing CNN/RNN with attention: The attention mechanisms can help us effectively shrink the parameter space, thus reducing most of the noise and unnecessary variability in training. Thus the stationary solutions are more likely to generalize well. That's why we start with analyzing the naive global attention model and self-attention models, which can help inspire the analysis of these complicated tasks. More detailed analysis/experiments with CNN/RNN are important future work. To this point, our experiments aim at validating the theoretical analyses and explaining why attention works in general.

## E PROOFS

### E.1 PROOF OF THEOREM 1

*Proof.* The proof is divided into two parts. Firstly, we study the landscape of population risk in part (a), then we evaluate the convergence of empirical risk to the population risk in part (b).

**a. Landscape of population risk**

We introduce necessary notations beforehand. To emphasize the role of $\boldsymbol{x}$ and $\boldsymbol{y}$ separately, here we denote $R(\boldsymbol{w}^{(1)}, \boldsymbol{w}^{(2)}, \boldsymbol{a}) = \mathbb{E}_{y|x}(R_n(\boldsymbol{w}^{(1)}, \boldsymbol{w}^{(2)}, \boldsymbol{a}))$, which is the expectation of the empirical loss gradient with respect to $\boldsymbol{y}$, treating $\boldsymbol{x}$ as random, and $\nabla R(\boldsymbol{w})$ as corresponding derivatives. And we denote $\mathbb{E}_x(\nabla R(\boldsymbol{w}^{(1)}, \boldsymbol{w}^{(2)}, \boldsymbol{a})) = \mathbb{E}_{x,y}(R_n(\boldsymbol{w}^{(1)}, \boldsymbol{w}^{(2)}, \boldsymbol{a}))$, which is the expectation of the empirical loss function with expectation to both $\boldsymbol{x}$ and $\boldsymbol{y}$. In our analysis, first we will study $\mathbb{E}_x(\nabla R(\boldsymbol{w}^{(1)}, \boldsymbol{w}^{(2)}, \boldsymbol{a}))$, then we analyze $\mathbb{E}_{y|x} R_n(\boldsymbol{w}^{(1)}, \boldsymbol{w}^{(2)}, \boldsymbol{a})$. The motivation of using $\mathbb{E}_{y|x} R_n(\boldsymbol{w}^{(1)}, \boldsymbol{w}^{(2)}, \boldsymbol{a})$ comes from that it simplifies the part (b) analysis of empirical risk convergence, since the randomness of $\boldsymbol{x}$ has been included in the population risk analysis, making advantage that the noise is independent from predictors. In the proof, we may use $o(\gamma)$ for vector/matrix case. In these cases, it means that every element in vector/matrix is $o(\gamma)$. In the proof, we will regard $d$ as an arbitrary large fixed value, not diverging with $n$.

We denote
$$u = (\boldsymbol{w}^{(2)} \phi(\langle \boldsymbol{w}^{(1)}, \boldsymbol{x} \odot \boldsymbol{a} \rangle) - \mathbb{E}(y|\boldsymbol{x})) \tag{7}$$

and $u_i$ as the version with specified sample index. Then the derivatives of population risk with expection to $\boldsymbol{y}$ can be presented as follows:

$$\nabla R(\boldsymbol{w}^{(2)}) = \frac{1}{n} \sum_{i=1}^{n} u_i \phi(\langle \boldsymbol{w}^{(1)}, \boldsymbol{x}_i \odot \boldsymbol{a} \rangle)$$

$$\nabla R(\boldsymbol{w}^{(1)}) = \frac{1}{n} \sum_{i=1}^{n} u_i (\boldsymbol{x}_i \odot \boldsymbol{a})(\boldsymbol{w}^{(2)} \odot \phi'(\langle \boldsymbol{w}^{(1)}, \boldsymbol{x}_i \odot \boldsymbol{a} \rangle))^T$$

$$\nabla R(\boldsymbol{a}) = \frac{1}{n} \sum_{i=1}^{n} u_i (\boldsymbol{x}_i \odot (\boldsymbol{w}^{(1)T}(\boldsymbol{w}^{(2)} \odot \phi'(\langle \boldsymbol{w}^{(1)}, \boldsymbol{x}_i \odot \boldsymbol{a} \rangle))))$$

By (A3), we know that $\mathbb{E}(y_i|\boldsymbol{x}_i) = \boldsymbol{w}^{(2)\star T}\phi(\langle \boldsymbol{w}^{(1)\star}, \boldsymbol{x}_i \odot \boldsymbol{a}^\star\rangle)$. Therefore when $(\boldsymbol{a}, \boldsymbol{w}^{(1)}, \boldsymbol{w}^{(2)}) = (\boldsymbol{a}^\star, \boldsymbol{w}^{(1)\star}, \boldsymbol{w}^{(2)\star})$, all the $u_i$ are zero, and all the gradients expectations are zero. Thus for any true set of parameter $(\boldsymbol{a}^\star, \boldsymbol{w}^{(1)\star}, \boldsymbol{w}^{(2)\star})$, they have zero gradient expectation automatically. And the key of our proof is showing that with high probability, any parameter $(\boldsymbol{a}, \boldsymbol{w}^{(1)}, \boldsymbol{w}^{(2)})$ cannot be stationary point if $\mathbb{E}(|\tilde{\boldsymbol{w}}^{(2)}\phi(\langle \tilde{\boldsymbol{w}}^{(1)}, \boldsymbol{x} \odot \tilde{\boldsymbol{a}}\rangle) - \mathbb{E}(y|\boldsymbol{x})|^2) \geq O(\gamma^2)$, because their gradients w.r.t to $\boldsymbol{w}^{(2)}$ or $\boldsymbol{w}^{(1)}$ must be bounded away from zero.

By our assumption (A2) and (A4), our parameters $\boldsymbol{w}^{(1)}, \boldsymbol{w}^{(2)}$ and $\boldsymbol{a}$ are inside the $\ell_2$ balls $B^d(0, C_1), B^{p\times d}(0, C_2)$ and $B^p(0, s_0^2)$. By Lemma 5.2 in Vershynin (2010), we know the $\epsilon$-covering number $N_{\epsilon_1}, N_{\epsilon_2}, N_{\epsilon_3}$ for these three balls are upper bounded by:

$$N_{\epsilon_1} \leq (3C_1/\epsilon)^d, N_{\epsilon_2} \leq (3C_2/\epsilon)^{pd}, N_{\epsilon_3} \leq (3s_0^2/\epsilon)^p$$

Then we know $3\epsilon$-covering number for the union of all three parameters $N_{3\epsilon}$ satisfies that $N_{3\epsilon} \leq N_{\epsilon_1}N_{\epsilon_2}N_{\epsilon_3}$. For the ease of notation, we denote $\boldsymbol{\theta} = (\boldsymbol{a}, \boldsymbol{w}^{(1)}, \boldsymbol{w}^{(2)})$. Let $\Theta_\epsilon = \{\boldsymbol{\theta}_1, \cdots, \boldsymbol{\theta}_{N_\epsilon}\}$ be a corresponding cover with $N_{3\epsilon}$ elements. Then we can always find $\Theta_\epsilon$ such that for any feasible $\boldsymbol{\theta}$, there exists $j \in [N]$ such that $\max(\|\boldsymbol{w}_{(j)}^{(1)} - \boldsymbol{w}^{(1)}\|_2, \|\boldsymbol{w}_{(j)}^{(2)} - \boldsymbol{w}^{(2)}\|_2, \|\boldsymbol{a}_{(j)} - \boldsymbol{a}\|_2) \leq \epsilon$. In this proof, we use parenthesis subscription $(j)$ to represent elements in the cover, to distinguish it from other subscriptions.

By triangle inequality, we have for $v = 1, 2$:

$$\|\nabla R(\boldsymbol{w}^{(v)})\|_2 \geq \|\nabla R(\boldsymbol{w}_{(j)}^{(v)})\|_2 - \|\nabla R(\boldsymbol{w}^{(v)}) - \nabla R(\boldsymbol{w}_{(j)}^{(v)})\|_2 \tag{8}$$

And in the following section, we prove that if $\mathbb{E}(u^2) \geq O(\gamma^2)$, we must have $\|\mathbb{E}_x(\nabla R(\boldsymbol{w}^{(2)}))\|_2 \geq O(\gamma)$ or there exists $k^{th}$ variable of $\boldsymbol{x}$, such that the $k^{th}$ column of $\nabla R(\boldsymbol{w}^{(1)})$ satisfies $\|\mathbb{E}_x(\nabla R(\boldsymbol{w}_k^{(1)}))\|_2 \geq O(\gamma)$. Here the subscription $k$ without parenthesis corresponds to $k^{th}$ feature.

First we consider $\nabla R(\boldsymbol{w}^{(2)})$. By (A4), there are at most $s_0$ nonzero elements in $\boldsymbol{x} \odot \boldsymbol{a}$, and we denote $\|\cdot\|_{1, active}$ as the $\ell_1$ norm on the elements with corresponding non-zero attention mask. Thus by inequality between norms, we have:

$$\mathbb{E}(\|\phi(\langle \boldsymbol{w}^{(1)}, \boldsymbol{x}_i \odot \boldsymbol{a}\rangle)\|_2) \leq |\boldsymbol{x} \odot \boldsymbol{a}|_{\max}|\boldsymbol{w}^{(1)}|_{1, active} \leq \sqrt{s_0}C_xC_1 = O(\sqrt{s_0})$$

Then we derive

$$\|\mathbb{E}_x(\nabla R(\boldsymbol{w}^{(2)}))\|_2 = \|cov(u, \phi(\langle \boldsymbol{w}^{(1)}, \boldsymbol{x} \odot \boldsymbol{a}\rangle)) + \mathbb{E}(u)\mathbb{E}(\phi(\langle \boldsymbol{w}^{(1)}, \boldsymbol{x} \odot \boldsymbol{a}\rangle))\|_2$$
$$\geq \|cov(u, \phi(\langle \boldsymbol{w}^{(1)}, \boldsymbol{x} \odot \boldsymbol{a}\rangle))\|_2 - o(\gamma/\sqrt{s_0})O(\sqrt{s_0})$$

Therefore if $\|cov(u, \phi(\langle \boldsymbol{w}^{(1)}, \boldsymbol{x}_i \odot \boldsymbol{a}\rangle))\|_2 \geq O(\gamma)$, we already have $\|\mathbb{E}(\nabla R(\boldsymbol{w}^{(2)}))\|_2 \geq O(\gamma)$. Then we consider the case when $\|cov(u, \phi(\langle \boldsymbol{w}^{(1)}, \boldsymbol{x}_i \odot \boldsymbol{a}\rangle))\|_2 = o(\gamma)$. In this case, we denote $\boldsymbol{r} = cov(\mathbb{E}(y|\boldsymbol{x}), \phi(\langle \boldsymbol{w}^{(1)}, \boldsymbol{x} \odot \boldsymbol{a}\rangle)) \in \mathbb{R}^d$, and denote the covariance matrix for $\phi(\langle \boldsymbol{w}^{(1)}, \boldsymbol{x} \odot \boldsymbol{a}\rangle)$ as:

$$(\boldsymbol{\Sigma}_\phi)_{ij} = cov(\phi_i(\langle \boldsymbol{w}^{(1)}, \boldsymbol{x} \odot \boldsymbol{a}\rangle), \phi_j(\langle \boldsymbol{w}^{(1)}, \boldsymbol{x} \odot \boldsymbol{a}\rangle))$$

Then plugging into $cov((\boldsymbol{w}^{(2)})^T\phi(\langle \boldsymbol{w}^{(1)}, \boldsymbol{x} \odot \boldsymbol{a}\rangle), \phi(\langle \boldsymbol{w}^{(1)}, \boldsymbol{x} \odot \boldsymbol{a}\rangle)) = \boldsymbol{r} + o(\gamma)$, using subtraction and addition, we have:

$$\boldsymbol{\Sigma}_\phi \boldsymbol{w}^{(2)} = \boldsymbol{r} + o(\gamma)$$

With (A5), covariance matrix $\boldsymbol{\Sigma}_\phi$ is invertible with smallest eigenvalue lower bounded, we have $\boldsymbol{w}^{(2)} = \boldsymbol{\Sigma}_\phi^{-1}\boldsymbol{r} + o(\gamma)$.

Next we argue the following term by contradiction:

$$\mathbb{E}(\|\phi(\langle \boldsymbol{w}^{(1)}, \boldsymbol{x} \odot \boldsymbol{a}\rangle) - \phi^\star(\langle \boldsymbol{w}^{(1)}, \boldsymbol{x} \odot \boldsymbol{a}\rangle)\|_2) \geq O(\gamma) \tag{9}$$

where $\phi^\star(\langle \boldsymbol{w}^{(1)}, \boldsymbol{x} \odot \boldsymbol{a}\rangle)$ represents the function corresponding to the true parameter set $(\boldsymbol{w}^{(1)\star}, \boldsymbol{w}^{(2)\star}, \boldsymbol{a}^\star)$. If we have equation 9 violated, we know:

$\boldsymbol{\Sigma}_{\phi_{ij}} - \boldsymbol{\Sigma}_{\phi_{ij}^\star} =$

$\mathbb{E}(\phi(\langle \boldsymbol{w}^{(1)}, \boldsymbol{x} \odot \boldsymbol{a}\rangle)_i\phi(\langle \boldsymbol{w}^{(1)}, \boldsymbol{x} \odot \boldsymbol{a}\rangle)_j) - \mathbb{E}(\phi(\langle \boldsymbol{w}^{(1)}, \boldsymbol{x} \odot \boldsymbol{a}\rangle)_i)\mathbb{E}(\phi(\langle \boldsymbol{w}^{(1)}, \boldsymbol{x} \odot \boldsymbol{a}\rangle)_j)$

$- (\mathbb{E}(\phi^\star(\langle \boldsymbol{w}^{(1)}, \boldsymbol{x} \odot \boldsymbol{a}\rangle)_i\phi^\star(\langle \boldsymbol{w}^{(1)}, \boldsymbol{x} \odot \boldsymbol{a}\rangle)_j) - \mathbb{E}(\phi^\star(\langle \boldsymbol{w}^{(1)}, \boldsymbol{x} \odot \boldsymbol{a}\rangle)_i)\mathbb{E}(\phi^\star(\langle \boldsymbol{w}^{(1)}, \boldsymbol{x} \odot \boldsymbol{a}\rangle)_j))$

$\lesssim O(1)(\mathbb{E}(\|\phi(\langle \boldsymbol{w}^{(1)}, \boldsymbol{x} \odot \boldsymbol{a}\rangle)_i - \phi^\star(\langle \boldsymbol{w}^{(1)}, \boldsymbol{x} \odot \boldsymbol{a}\rangle_i\|_2 + \mathbb{E}(\|\phi(\langle \boldsymbol{w}^{(1)}, \boldsymbol{x} \odot \boldsymbol{a}\rangle)_j - \phi^\star(\langle \boldsymbol{w}^{(1)}, \boldsymbol{x} \odot \boldsymbol{a}\rangle_j\|_2))$

$= o(\gamma)$

where we use $\mathbb{E}(\|\phi(\langle \boldsymbol{w}^{(1)}, \boldsymbol{x} \odot \boldsymbol{a}\rangle)\|_2) \leq \mathbb{E}(\|\phi^\star(\langle \boldsymbol{w}^{(1)}, \boldsymbol{x} \odot \boldsymbol{a}\rangle)\|_2) + \mathbb{E}(\|\phi(\langle \boldsymbol{w}^{(1)}, \boldsymbol{x} \odot \boldsymbol{a}\rangle) - \phi^\star(\langle \boldsymbol{w}^{(1)}, \boldsymbol{x} \odot \boldsymbol{a}\rangle)\|_2) = O(1)$, and we know $\phi^\star(\langle \boldsymbol{w}^{(1)}, \boldsymbol{x} \odot \boldsymbol{a}\rangle)$ corresponding to true parameter set is finite. Since we have derived $\boldsymbol{w}^{(2)\star} = \boldsymbol{\Sigma}_{\phi^\star}^{-1} \boldsymbol{r}$, with lower bounded eigenvalue assumption, we can derive:

$$\boldsymbol{\Sigma}_\phi^{-1} - \boldsymbol{\Sigma}_{\phi^\star}^{-1} = \boldsymbol{\Sigma}_\phi^{-1} \boldsymbol{\Sigma}_\phi (\boldsymbol{\Sigma}_\phi^{-1} - \boldsymbol{\Sigma}_{\phi^\star}^{-1}) = \boldsymbol{\Sigma}_\phi^{-1} (\boldsymbol{I} - (\boldsymbol{\Sigma}_{\phi^\star} + o(\gamma))\boldsymbol{\Sigma}_{\phi^\star}^{-1})$$
$$= \boldsymbol{\Sigma}_\phi^{-1} o(\gamma) \boldsymbol{\Sigma}_{\phi^\star}^{-1} = o(\gamma)$$

We know if we have $\|cov(u, \phi(\langle \boldsymbol{w}^{(1)}, \boldsymbol{x}_i \odot \boldsymbol{a}\rangle))\|_2 = o(\gamma)$, thus $\boldsymbol{w}^{(2)} = \boldsymbol{\Sigma}_\phi^{-1} \boldsymbol{r} + o(\gamma)$. Therefore we conclude $\boldsymbol{w}^{(2)} = \boldsymbol{w}^{(2)\star} + o(\gamma)$. Plugging back to the formula of $u$ in equation 12, we have:

$$\mathbb{E}(u^2) = \mathbb{E}(|\boldsymbol{w}^{(2)\star T} \phi(\boldsymbol{w}^{(1)\star}, \boldsymbol{x} \odot \boldsymbol{a}) - \boldsymbol{w}^{(2)T} \phi(\langle \boldsymbol{w}^{(1)}, \boldsymbol{x}_i \odot \boldsymbol{a}\rangle)|^2)$$
$$\leq \mathbb{E}(|\boldsymbol{w}^{(2)\star T} \phi(\boldsymbol{w}^{(1)\star}, \boldsymbol{x} \odot \boldsymbol{a}) - \boldsymbol{w}^{(2)\star T} \phi(\langle \boldsymbol{w}^{(1)}, \boldsymbol{x}_i \odot \boldsymbol{a}\rangle)|^2)$$
$$+ \mathbb{E}(|\boldsymbol{w}^{(2)\star T} \phi(\langle \boldsymbol{w}^{(1)}, \boldsymbol{x}_i \odot \boldsymbol{a}\rangle) - \boldsymbol{w}^{(2)T} \phi(\langle \boldsymbol{w}^{(1)}, \boldsymbol{x}_i \odot \boldsymbol{a}\rangle)|^2) = o(\gamma^2)$$

Here we conclude the contradiction. When condition equation 9 is violated, we derive $\mathbb{E}(u^2) = o(\gamma^2)$. Therefore if $\mathbb{E}(u^2) \geq O(\gamma^2)$, we must have $\mathbb{E}(\|\phi(\langle \boldsymbol{w}^{(1)}, \boldsymbol{x}_i \odot \boldsymbol{a}\rangle) - \phi^\star(\langle \boldsymbol{w}^{(1)} \boldsymbol{x}_i \odot \boldsymbol{a}\rangle\|_2) \geq O(\gamma)$.

Now we are ready to study $\nabla R(\boldsymbol{w}^{(1)})$. Recall in assumption (A5), we denote $h_k(\boldsymbol{x}) = x_{\cdot k} a_k (\boldsymbol{w}^{(2)} \odot \phi'(\langle \boldsymbol{w}^{(1)}, \boldsymbol{x}_i \odot \boldsymbol{a}\rangle))$, where $x_{\cdot k}$ represents $k^{th}$ feature in $\boldsymbol{x}$, and $a_k$ is the attention weight for $x_{\cdot k}$. And we have proved that $\mathbb{E}(\|\phi(\langle \boldsymbol{w}^{(1)}, \boldsymbol{x}_i \odot \boldsymbol{a}\rangle) - \phi^\star(\langle \boldsymbol{w}^{(1)} \boldsymbol{x}_i \odot \boldsymbol{a}\rangle\|_2) = O(\gamma)$. Thus by (A5), we can find $k, t$ such that $sd(h_t(x_{\cdot k})) = O(1)$ and $cor(u, h_t(x_{\cdot k})) = O(1)$. Then we have $|cov(u, h_t(x_{\cdot k}))| = sd(u)sd(h_t(x_{\cdot k}))|cor(u, h_t(x_{\cdot k}))|$. We know

$$sd(u) = \sqrt{\mathbb{E}(u^2) - |\mathbb{E}(u)|^2} = \sqrt{O(\gamma^2) - o(\gamma)^2} = O(\gamma)$$

Therefore we have:

$$\|\mathbb{E}(\nabla R(\boldsymbol{w}_k^{(1)}))\|_2 \geq \|\mathbb{E}(u h_t(x_{\cdot k}))\|_2 = \|cov(u, h_t(x_{\cdot k}))) + \mathbb{E}(u)\mathbb{E}(h_t(x_{\cdot k}))\|_2 = O(\gamma)$$

Here we conclude that we can always find $k \in \{1, \ldots, p\}$, such that $\|\mathbb{E}(\nabla R(\boldsymbol{w}_k^{(1)}))\|_2 \geq O(\gamma)$.

With this conclusion, we move to bound the gradient term $\boldsymbol{v}_i = u_i \phi(\langle \boldsymbol{w}^{(1)}, \boldsymbol{x}_i \odot \boldsymbol{a}\rangle)$ and $\boldsymbol{z}_{ik} = u_i (\boldsymbol{x}_{ik} \odot a_k)(\boldsymbol{w}^{(2)T}) \odot \phi'(\langle \boldsymbol{w}^{(1)}, \boldsymbol{x}_i \odot \boldsymbol{a}\rangle))$. With any fixed parameter set, we can calculate:

$$\|\boldsymbol{v}_i\|_2^2 \lesssim (C_2 \|\phi(\langle \boldsymbol{w}^{(1)}, \boldsymbol{x}_i \odot \boldsymbol{a}\rangle)\|_2)^2 \|\phi(\langle \boldsymbol{w}^{(1)}, \boldsymbol{x}_i \odot \boldsymbol{a}\rangle)\|_2^2 = s_0^2 C_1^4 C_2^2 C_x^4$$
$$\|\boldsymbol{z}_{ik}\|_2^2 \lesssim (C_2 \|\phi(\langle \boldsymbol{w}^{(1)}, \boldsymbol{x}_i \odot \boldsymbol{a}\rangle)\|_2)^2 C_x^2 C_2^2 = s_0 C_1^2 C_2^4 C_x^2$$

where we use again that there are at most $s_0$ nonzero elements in $\boldsymbol{x} \odot \boldsymbol{a}$:

$$\|\phi(\langle \boldsymbol{w}^{(1)}, \boldsymbol{x}_i \odot \boldsymbol{a}\rangle)\|_2 \leq \max\{|\boldsymbol{x} \odot \boldsymbol{a}|\}\|\boldsymbol{w}^{(1)}\|_{1, active} = \sqrt{s_0} C_x C_1$$

From the last section, we know there exists a constant $c$ such that $\|\mathbb{E}_x(\nabla R(\boldsymbol{w}^{(2)}))\|_2 \geq c\gamma$ or $\|\mathbb{E}_x(\nabla R(\boldsymbol{w}_k^{(1)}))\|_2 \geq c\gamma$ for some constant c. Suppose $\|\mathbb{E}_x(\nabla R(\boldsymbol{w}^{(2)}))\|_2 \geq c\gamma$, and we denote $\sigma^2 = s_0^2 C_1^4 C_2^2 C_x^4$, And we know $\|\boldsymbol{v}_i\|_2^2$ is bounded as we derived. Therefore we can use Hoeffding bound on the $\ell_2$ norm in direction of $\mathbb{E}_x(\nabla R(\boldsymbol{w}_{(j)}^{(2)}))$, since we know the variance on this direction is smaller than $\sigma^2$. Denoting $\nabla R(\boldsymbol{w}_{(j)}^{(2)})$ as the gradient with respect to $j^{th}$ parameter set in $\epsilon$-cover for $j \in \{1, \ldots, N_\epsilon\}$:

$$P(\|\nabla R(\boldsymbol{w}_{(j)}^{(2)}) - \mathbb{E}_x(\nabla R(\boldsymbol{w}_{(j)}^{(2)}))\|_2 \geq \frac{c\gamma}{3}) \lesssim \exp(-n \frac{c^2 \gamma^2}{\sigma^2})$$

By union bound, we have:

$$P(\exists j \in [N_\epsilon], \|\nabla R(\boldsymbol{w}_{(j)}^{(2)})\|_2 \geq \frac{2c\gamma}{3}) \lesssim N_\epsilon \exp(-n \frac{c^2 \gamma^2}{\sigma^2})$$

Secondly we analyze $\|\nabla R(\boldsymbol{w}^{(2)}) - \nabla R(\boldsymbol{w}_{(j)}^{(2)})\|_2$ term. Here we use $u_i$ to represent the prediction error for $i^{th}$ instant with respect to parameter $(\boldsymbol{a}, \boldsymbol{w}^{(1)}, \boldsymbol{w}^{(2)})$, and use $u_{i(j)}$ to represent the term

with respect to the parameter from $j^{th}$ element in $\epsilon$-cover set. By triangle inequality, we have:

$$\|\nabla R(\boldsymbol{w}^{(2)}) - \nabla R(\boldsymbol{w}^{(2)}_{(j)})\|_2 \leq \frac{2}{n} \|\sum_{i=1}^{n} (u_i \phi(\langle \boldsymbol{w}^{(1)}, \boldsymbol{x}_i \odot \boldsymbol{a}\rangle) - u_{i(j)} \phi(\langle \boldsymbol{w}^{(1)}_{(j)}, \boldsymbol{x}_i \odot \boldsymbol{a}_{(j)}\rangle))\|_2$$

$$\lesssim \frac{2}{n} (\|\sum_{i=1}^{n} (u_i - u_{i(j)}) \phi(\langle \boldsymbol{w}^{(1)}, \boldsymbol{x}_i \odot \boldsymbol{a}\rangle)\|_2 + \|\sum_{i=1}^{n} u_{i(j)} (\phi(\langle \boldsymbol{w}^{(1)}, \boldsymbol{x}_i \odot \boldsymbol{a}\rangle) - \phi(\langle \boldsymbol{w}^{(1)}_{(j)}, \boldsymbol{x}_i \odot \boldsymbol{a}_{(j)}\rangle))\|_2)$$

$$\lesssim s_0 C_X^2 C_1^2 C_2 \epsilon$$

We choose $\epsilon = \frac{c\gamma}{3 s_0 C_x^2 C_1^2 C_2}$, and plug back above results to equation 8, then at least with probability $1 - \mathcal{O}(N_\epsilon \exp(-n\frac{c^2\gamma^2}{\sigma^2}))$, we have $\|\nabla R(\boldsymbol{w}^{(2)})\|_2 > \frac{c\gamma}{3}$. Therefore we can choose $n \gtrsim \frac{\sigma^2}{c^2\gamma^2} \log(\frac{s_0 C_1 C_2 C_x}{c\gamma})(pd + p + d)$, such that $N_\epsilon \exp(-n\frac{c^2\gamma^2}{\sigma^2}) = o(1)$. Finally we can conclude that with probability $1 - o_n(1)$, for any $(\boldsymbol{a}, \boldsymbol{w}^{(1)}, \boldsymbol{w}^{(2)})$ such that $\mathbb{E}(\tilde{\boldsymbol{w}}^{(2)} \phi(\langle \tilde{\boldsymbol{w}}^{(1)}, \boldsymbol{x} \odot \tilde{\boldsymbol{a}}\rangle) - \mathbb{E}(y|\boldsymbol{x}))^2 \geq \gamma$, we have $\|\nabla R(\boldsymbol{w}^{(2)})\|_2 > \frac{c\gamma}{3}$.

If $\|\mathbb{E}_x(\nabla R(\boldsymbol{w}^{(1)}_k))\|_2 = O(\gamma)$. Applying the same technique, we can show that with probability $1 - o_n(1)$, we have $\|\nabla R(\boldsymbol{w}^{(1)}_k)\|_2 > \frac{c\gamma}{3}$.

## b. Convergence of empirical risk

So far, we have shown that for population risk with respect to $y$, with high probability, all the parameter sets with poor prediction in expectation, i.e $\mathbb{E}(|\tilde{\boldsymbol{w}}^{(2)} \phi(\langle \tilde{\boldsymbol{w}}^{(1)}, \boldsymbol{X} \odot \tilde{\boldsymbol{a}}\rangle) - \mathbb{E}(y|\boldsymbol{x})|^2) \geq O(\gamma^2)$, their population risk gradient with expectation to $\boldsymbol{y}$ must be away from zero. Now we move forward to show that empirical risk will converge to the popular risk, i.e. $\nabla R_n(\boldsymbol{w}) \to \nabla R(\boldsymbol{w})$. Thus these parameter sets cannot have zero empirical gradient. In aspect of three parameter sets, they can be represented as:

$$\nabla R_n(\boldsymbol{w}^{(2)}) - \nabla R(\boldsymbol{w}^{(2)}) = \frac{1}{n}\sum_{i=1}^{n}\epsilon_i \phi(\langle \boldsymbol{w}^{(1)}, \boldsymbol{x}_i \odot \boldsymbol{a}\rangle)$$

$$\nabla R_n(\boldsymbol{w}^{(1)}) - \nabla R(\boldsymbol{w}^{(1)}) = \frac{1}{n}\sum_{i=1}^{n}\epsilon_i (\boldsymbol{x}_i \odot \boldsymbol{a})(\boldsymbol{w}^{(2)} \odot \phi'(\langle \boldsymbol{w}^{(1)}, \boldsymbol{x}_i \odot \boldsymbol{a}\rangle))^T$$

$$\nabla R_n(\boldsymbol{a}) - \nabla R(\boldsymbol{a}) = \frac{1}{n}\sum_{i=1}^{n}\epsilon_i (\boldsymbol{x}_i \odot (\boldsymbol{w}^{(1)T}(\boldsymbol{w}^{(2)} \odot \phi'(\langle \boldsymbol{w}^{(1)}, \boldsymbol{x}_i \odot \boldsymbol{a}\rangle))))$$

With (A3), we know that $\epsilon_i \sim subG(0, C_4^2)$, thus $\frac{1}{n}\sum_{i=1}^{n}\epsilon_i = O(\frac{1}{\sqrt{n}})$ by C.L.T, combining the bound for $\phi(\langle \boldsymbol{w}^{(1)}, \boldsymbol{x}_i \odot \boldsymbol{a}\rangle)$ we have derived in last section, with sample size $n \gtrsim \frac{\sigma^2}{c^2\gamma^2} \log(\frac{s_0 C_1 C_2 C_x}{c\gamma})(pd + p + d)$, conclude that with probability $1 - o_p(1)$:

$$\|\nabla R_n(\boldsymbol{w}^{(2)}) - \nabla R(\boldsymbol{w}^{(2)})\|_2 \leq \frac{c\gamma}{6} \tag{10}$$

$$\|\nabla R_n(\boldsymbol{w}^{(1)}_k) - \nabla R(\boldsymbol{w}^{(1)}_k)\|_2 \leq \frac{c\gamma}{6} \tag{11}$$

Recalling part (a), under the first case that w.h.p $\|\nabla R(\boldsymbol{w}^{(2)})\|_2 \geq \frac{c\gamma}{3}$ for any parameter $(\boldsymbol{a}, \boldsymbol{w}^{(1)}, \boldsymbol{w}^{(2)})$ with $\|\tilde{\boldsymbol{w}}^{(2)} \phi(\langle \tilde{\boldsymbol{w}}^{(1)}, \boldsymbol{X} \odot \tilde{\boldsymbol{a}}\rangle) - \mathbb{E}(\boldsymbol{y}|\boldsymbol{X})\|_2 \geq \gamma$. Combining this with (10), we can conclude that for any positive constant $\gamma > 0$, with required sample size, with high probability that $\|\nabla R_n(\boldsymbol{w}^{(2)})\|_2 > 0$, thus they cannot be stationary solution for our loss function. As we stated, if we are in another case, we have w.h.p $\|\nabla R(\boldsymbol{w}^{(1)}_k)\|_2 \geq \frac{c\gamma}{3}$, we can use same techniques to show w.h.p $\|\nabla R_n(\boldsymbol{w}^{(2)})\|_2 > 0$.

In other words, under our assumptions, all the stationary points $(\tilde{\boldsymbol{a}}, \tilde{\boldsymbol{w}}^{(1)}, \tilde{\boldsymbol{w}}^{(2)})$ in our programming satisfy the prediction error upper bound rate $\gamma$ w.h.p. $\square$

### E.2 PROOF OF COROLLARY 1

*Proof.* To extend the result from Theorem 3 to corollary 1, we simply substitute the $\ell_2$ norm bound $\|\boldsymbol{x}_i \odot \boldsymbol{a}\|_2$, from $s_0 C_x$ to $C_x$ since $\|\boldsymbol{a}\|_2 \le \|\boldsymbol{a}\|_1 = 1$. All the other parts keep the same. Thus the only difference is that we remove $s_0$ in the bound comparing with Theorem 3. $\square$

### E.3 PROOF OF PROPOSITION 1

*Proof.* This proposition is a direct result of Theorem 3. Since the assumptions for $\boldsymbol{a}$ still hold, all the bounds apply. The only different is that since $\boldsymbol{a}$ is not optimized together, we don't have to consider the $\epsilon$-cover number for $\boldsymbol{a}$ in the maximum operator. This leads to a slightly tighter sample complexity bound in corollary 1 comparing with theorem 3. $\square$

### E.4 PROOF OF THEOREM 2

*Proof.* Similar with Theorem 1, we obtained a new $\epsilon$-covering bound:

$$N_{\epsilon_1} \le (3C_1/\epsilon)^d, N_{\epsilon_2} \le (3C_2/\epsilon)^{pd_v d}, N_{\epsilon_3} \le (3C_5/\epsilon)^{td_q}, N_{\epsilon_4} \le (3C_6/\epsilon)^{td_q}, N_{\epsilon_5} \le (3C_7/\epsilon)^{td_v},$$

where $N = \Pi_{i=1}^5 N_{\epsilon_i}$. And also, the new $\boldsymbol{v}_i$ and $\boldsymbol{z}_{ik}$ terms are:

$$\boldsymbol{v}_i = \boldsymbol{u}_i \phi(\langle \boldsymbol{w}^{(1)}, vec(\boldsymbol{w}^V \boldsymbol{x}_i \boldsymbol{a}_i^{self}) \rangle)$$

$$\boldsymbol{z}_{ik} = u_i (\boldsymbol{w}^V \boldsymbol{x}_i \boldsymbol{a}_i^{self})_{.k} (\boldsymbol{w}^{(2T)}) \odot \phi^{'}(\langle \boldsymbol{w}^{(1)}, vec(\boldsymbol{w}^V \boldsymbol{x}_i \boldsymbol{a}_i^{self}) \rangle)))$$

where $\boldsymbol{u}_i = y_i - (\boldsymbol{w}^{(2T)} \phi(\langle \boldsymbol{w}^{(1)}, vec(\boldsymbol{w}^V \boldsymbol{x}_i \boldsymbol{a}_i^{self}))$. Under assumptions, using the same agrument with Theorem 3, we can show there exists a constant $c$ such that either $\|\mathbb{E}(\nabla R(\boldsymbol{w}^{(2)}))\|_2 \ge c\gamma$ or there exist $k \in 1, \dots, p$ such that $\|\mathbb{E}(\nabla R(\boldsymbol{w}_k^{(1)}))\|_2 \ge c\gamma$.

In the case when $\|\mathbb{E}(\nabla R(\boldsymbol{w}^{(2)}))\|_2 \ge c\gamma$. we have new bound of $\|\boldsymbol{v}\|_2^2$ with respect to $\boldsymbol{x}$ is upper bounded by $\sigma^2 = p^2 t^2 C_1^4 C_2^2 C_7^4 C_X^4$, considering $\boldsymbol{a}^{self}$ is normalized by softmax function for each vector in set. Parallel to Theorem 3, by hoeffding bound and union bound, we have:

$$P(\exists j \in [N_\epsilon], \|\boldsymbol{v}_j\|_2 \le \frac{2c\gamma}{3}) \lesssim N_\epsilon \exp(-n\frac{c^2\gamma^2}{\sigma^2})$$

Secondly we bound $\|\nabla R(\boldsymbol{w}^{(2)}) - \nabla R(\boldsymbol{w}^{(2)})_j\|_2$ term by subtraction and addition:

$$\|\nabla R(\boldsymbol{w}^{(2)}) - \nabla R(\boldsymbol{w}^{(2)})_j\|_2$$

$$\le \frac{1}{n} \| \sum_{i=1}^n u_i^j \phi(\langle \boldsymbol{w}^{(1)}, vec(\boldsymbol{w}^V \boldsymbol{x}_i \boldsymbol{a}_i^{self(z)}) \rangle) - u_{i(j)} \phi(\langle \boldsymbol{w}_j^{(1)}, vec(\boldsymbol{w}^V \boldsymbol{x}_i \boldsymbol{a}_i^{self(z)}) \rangle) \|_2$$

$$\lesssim \frac{1}{n}(\| \sum_{i=1}^n (u_i - u_{i(j)}) \phi(\langle \boldsymbol{w}^{(1)}, vec(\boldsymbol{w}^V \boldsymbol{x}_i \boldsymbol{a}_i^{self(z)}) ) \|_2$$

$$+ \| \sum_{i=1}^n u_i^j (\phi(\langle \boldsymbol{w}^{(1)}, \boldsymbol{w}^V \boldsymbol{x}_i \boldsymbol{a}_i^{self(z)}) - u_{i(j)} \phi(\langle \boldsymbol{w}_j^{(1)}, vec(\boldsymbol{w}_{(j)}^V \boldsymbol{x}_i \boldsymbol{a}_{i(j)}^{self(z)}))) \|_2)$$

$$\lesssim pt C_1^2 C_2^2 C_7^2 C_x^2 \epsilon$$

recalling that $u_{i(j)}$ and $a_{i(j)}$ are corresponding to the $j^{th}$ epsilon cover. We choose $\epsilon = \frac{c\gamma}{3C_1^2 C_2^2 C_7^2 C_x^2}$, and combine the above results. Then at least with probability $1 - O(N_\epsilon \exp(-n\frac{c^2 s\gamma^2}{\sigma^2}))$, we have $\|\nabla R(\boldsymbol{w}^{(2)})\|_2 > \frac{\gamma}{3}$. Therefore we can choose $n \gtrsim \frac{\sigma^2}{c^2\gamma^2} \log(\frac{pC_1 C_2 C_5 C_6 C_7 C_x}{c^3\gamma})(pd_v d + d + 2pd_q)$, such then $N_\epsilon \exp(-n\frac{c^2\gamma^2}{\sigma^2}) = o(1)$. Thus with this required sample complexity, we have $\|\nabla R(\boldsymbol{w}^{(2)}) - \mathbb{E}(\nabla R(\boldsymbol{w}^{(2)}))\|_2 \le \frac{2c\gamma}{3}$. In the same way, we can in the case when show $\|\nabla R(\boldsymbol{w}_k^{(1)}) - \mathbb{E}(\nabla R(\boldsymbol{w}_k^{(1)}))\|_2 \le \frac{2c\gamma}{3}$.

Finally we can conclude that with high probability, any parameter $(\boldsymbol{a}, \boldsymbol{w}^{(1)}, \boldsymbol{w}^{(2)})$ with $\mathbb{E}(\tilde{\boldsymbol{w}}^{(2)} \phi(\langle \tilde{\boldsymbol{w}}^{(1)}, vec(\boldsymbol{w}^V \boldsymbol{x}_i \boldsymbol{a}_i^{self(z)}))))) - \mathbb{E}(y|\boldsymbol{x}))^2 \ge \gamma$, we have $\|\nabla R(\boldsymbol{w}^{(2)})\|_2 > \frac{\gamma}{3}$ or $\|\nabla R(\boldsymbol{w}_k^{(1)})\|_2 > \frac{\gamma}{3}$. Then following the same empirical risk convergence argument, we show that with high probability they cannot be stationary point. $\square$

## E.5 PROOF OF THEOREM 3

*Proof.* First with $\|\boldsymbol{a}\|_0 = s_0$, we know all the inputs $\boldsymbol{x}_i$ with corresponding $\boldsymbol{a}_i = 0$, will be inactive in the network. We can omit all these inactive inputs. Then we split $n_1$ units into $s_0$ group, with $\lfloor \frac{n_1}{s_0} \rfloor$ number of units in each group, and discard the leftover units. $s_0$ different groups correspond to $s_0$ active inputs with non-zero attention weight.

Inside each group, for example in $j^{th}$ group, denoting $q = \lfloor \frac{n_1}{s_0} \rfloor$, we choose the input weights and biases for $i = 1, 2, \cdots, q$ as:

$$h_1(\boldsymbol{x}) = \max\{0, \boldsymbol{w}_j \boldsymbol{x}\},$$
$$h_2(\boldsymbol{x}) = \max\{0, 2\boldsymbol{w}_j \boldsymbol{x} - 1\},$$
$$\vdots$$
$$h_q(\boldsymbol{x}) = \max\{0, 2\boldsymbol{w}_j \boldsymbol{x} - (q-1)\}$$

here we assign $\boldsymbol{w}_j$ to be a row vector with $j^{th}$ variable equal to 1 and all other entries to be 0. And in the second layer, we choose $\boldsymbol{w}^{(2)} = (\boldsymbol{w}_3, \cdots, \boldsymbol{w}_3)$, where $\boldsymbol{w}_3 = (1, -1, 1, \cdots, (-1)^{q+1})$, corresponding to $h_1$ to $h_q$ in each group. Then the designed network has $q$ linear regions inside each group, giving by the intervals:

$$(-\infty, 0], (0, 1], (1, 2], \cdots, [q-1, \infty)$$

Each of these intervals has a subset that is mapped by $\boldsymbol{w}_3 h(\boldsymbol{x})$ onto the interval (0,1).Montufar et al. (2014) Therefore the total number of linear regions is lower bounded by $\lfloor \frac{n_1}{s_0} \rfloor^{s_0}$. $\square$

## E.6 PROOF OF THEOREM 4

*Proof.* Here we define an $(\alpha_1, \alpha_2)$ scale transformation such that:

$$T_{\alpha_1, \alpha_2} : (\boldsymbol{a}, \boldsymbol{w}^{(1)}, \boldsymbol{w}^{(2)}) \mapsto (\alpha_1 \boldsymbol{a}, \alpha_2 \boldsymbol{w}^{(1)}, (\alpha_1 \alpha_2)^{-1} \boldsymbol{w}^{(2)})$$

Then we know the jacobian determinant for $T_{\alpha_1, \alpha_2}$ is $\alpha_1^{p-d} \alpha_2^{pd-d}$. Let $r > 0$ such that $B_\infty(r, \boldsymbol{\theta})$ is in $C(L, \boldsymbol{\theta}, \epsilon)$ and has empty intersection with $(\boldsymbol{a}, \boldsymbol{w}^{(1)}, \boldsymbol{w}^{(2)}) = \boldsymbol{0}$. Since $pd > d$, we assign $\alpha_2 \to \infty$, such that the jacobian determinant goes to infinity, and the volume of $C(L, \boldsymbol{\theta}, \epsilon)$ goes to infinity.

For the Hessian matrix, without loss of generality, we assume there is a positive diagonal element $\delta > 0$ in $\boldsymbol{a}$. Therefore the Frobenius norm $\|\nabla^2 L(T_{\alpha_1, \alpha_2}(\boldsymbol{\theta}))\|_F$ of

$$\nabla^2 L(T_{\alpha_1, \alpha_2}(\boldsymbol{\theta})) =$$
$$\begin{bmatrix} \alpha_1^{-1}\boldsymbol{I} & 0 & 0 \\ 0 & \alpha_2^{-1}\boldsymbol{I} & 0 \\ 0 & 0 & (\alpha_1\alpha_2)\boldsymbol{I} \end{bmatrix} \nabla^2 L(\boldsymbol{\theta}) \begin{bmatrix} \alpha_1^{-1}\boldsymbol{I} & 0 & 0 \\ 0 & \alpha_2^{-1}\boldsymbol{I} & 0 \\ 0 & 0 & (\alpha_1\alpha_2)\boldsymbol{I} \end{bmatrix}$$

is lower bounded by $\alpha_1^{-2}\delta$. Further we apply the fact that the biggest eigenvalue of a symmetric matrix $\boldsymbol{X}$ is larger than $c\|\boldsymbol{X}\|_F$, and pick $\alpha_1 < \sqrt{\frac{c\delta}{M}}$, then we have the biggest eigenvalue of $\nabla^2 L(T_{\alpha_1, \alpha_2}(\boldsymbol{\theta}))$ is larger than M. Therefore there exists a stationary point such that the operator norm for Hessian is arbitrary large. Thus we finish proving part (a).

Then we consider part (b). Similar with part (a), we define an $\alpha$ scale transformation such that:

$$T_\alpha : (\boldsymbol{w}^{(1)}, \boldsymbol{w}^{(2)}) \mapsto (\alpha \boldsymbol{w}^{(1)}, \alpha^{-1} \boldsymbol{w}^{(2)})$$

And all the value,query and key matrices remain the same. Then we know the jacobian determinant for $T_\alpha = \alpha^{(pd_v-1)d}$. Since $pd_v d \geq d$, as we assign $\alpha \to \infty$, such that the jacobian determinant goes to infinity, and the volume of $C(L, \boldsymbol{\theta}, \epsilon)$ goes to infinity.

For the Hessian matrix, we still assume a positive diagonal element $\delta > 0$ in $\boldsymbol{w}^{(1)}$. Similarly we have the Frobenius norm $\|\nabla^2 L(T_\alpha(\boldsymbol{\theta}))\|_F$ of

$$\nabla^2 L(T_\alpha(\boldsymbol{\theta})) =$$

$$\begin{bmatrix} \alpha^{-1}\boldsymbol{I} & 0 & 0 \\ 0 & \alpha\boldsymbol{I} & 0 \\ 0 & 0 & \boldsymbol{I} \end{bmatrix} \nabla^2 L(\boldsymbol{\theta}) \begin{bmatrix} \alpha^{-1}\boldsymbol{I} & 0 & 0 \\ 0 & \alpha\boldsymbol{I} & 0 \\ 0 & 0 & \boldsymbol{I} \end{bmatrix}$$

is lower bounded by $\alpha^{-2}\delta$. When we choose sufficient small $\alpha$, we have the biggest eigenvalue of $\nabla^2 L(T_{\alpha_1,\alpha_2}(\boldsymbol{\theta}))$ is larger than any constant $M$. Therefore there exists a stationary point such that the operator norm for Hessian is arbitrary large. $\qquad\square$

### E.7 PROOF OF THEOREM 5

*Proof.* For global attention in part (a), We start with calculating the gradient of the empirical loss function $\nabla R_n(\boldsymbol{w}^{(2)})$, where $R_n(\boldsymbol{w}^{(1)}, \boldsymbol{w}^{(2)}, \boldsymbol{a}) = \frac{1}{2n}\sum_{i=1}^{n}(\boldsymbol{w}^{(2)T}\phi(\langle\boldsymbol{w}^{(1)}, \boldsymbol{x}_i \odot \boldsymbol{a}\rangle) - y_i)^2$. Denoting $u_i = (\boldsymbol{w}^{(2)T}\phi(\langle\boldsymbol{w}^{(1)}, \boldsymbol{x}_i \odot \boldsymbol{a}\rangle) - y)$. The derivatives can be presented as follows:

$$\nabla R_n(\boldsymbol{w}^{(2)}) = \frac{1}{n}\sum_{i=1}^{n} u_i\phi(\langle\boldsymbol{w}^{(1)}, \boldsymbol{x}_i \odot \boldsymbol{a}\rangle)$$

$$\nabla R_n(\boldsymbol{w}^{(1)}) = \frac{1}{n}\sum_{i=1}^{n} u_i(\boldsymbol{x}_i \odot \boldsymbol{a})(\boldsymbol{w}^{(2)} \odot \phi^{'}(\langle\boldsymbol{w}^{(1)}, \boldsymbol{x}_i \odot \boldsymbol{a}\rangle))^T$$

$$\nabla R_n(\boldsymbol{a}) = \frac{1}{n}\sum_{i=1}^{n} u_i(\boldsymbol{x}_i \odot (\boldsymbol{w}^{(1)T}(\boldsymbol{w}^{(2)} \odot \phi^{'}(\langle\boldsymbol{w}^{(1)}, \boldsymbol{x}_i \odot \boldsymbol{a}\rangle))))$$

By assumption, $rank(\phi(\langle\boldsymbol{w}^{(1)}, \boldsymbol{x}_i \odot \boldsymbol{a}\rangle)_{i=1,...,n}) = n$, thus solving the linear system, we must have $u_i = 0$ for any $i = 1, 2, ..., n$ to satisfy that $\nabla R_n(\tilde{\boldsymbol{w}}^{(2)}) = 0$. Thus we know that the loss is exactly zero inside sample. Thus it must be a global minimum.

Part (b) can be proved by substituting $\boldsymbol{a}$ to $\boldsymbol{a}$ in part (a). Here we only consider the derivatives with respect to $\boldsymbol{w}^{(1)}$ and $\boldsymbol{w}^{(2)}$, they can be presented as follows:

$$\nabla R_n(\boldsymbol{w}^{(2)}) = \frac{1}{n}\sum_{i=1}^{n} u_i\phi(\langle\boldsymbol{w}^{(1)}, vec(\boldsymbol{w}^V\boldsymbol{x}_i\boldsymbol{a}_i^{self})\rangle)$$

$$\nabla R_n(\boldsymbol{w}^{(1)}) = \frac{1}{n}\sum_{i=1}^{n} u_i(vec(\boldsymbol{w}^V\boldsymbol{x}_i\boldsymbol{a}_i^{self}))(\boldsymbol{w}^{(2)} \odot \phi^{'}(\langle\boldsymbol{w}^{(1)}, vec(\boldsymbol{w}^V\boldsymbol{x}_i\boldsymbol{a}_i^{self}))^T$$

By assumption, $rank(\phi(\langle\boldsymbol{w}^{(1)}, vec(\boldsymbol{w}^V\boldsymbol{x}_i\boldsymbol{a}_i^{self})\rangle)_{i=1,...,n}) = n$, thus solving the linear system, we must have $u_i = 0$ for any $i = 1, 2, ..., n$ to satisfy that $\nabla R_n(\tilde{\boldsymbol{w}}^{(2)}) = 0$. Thus we know that the loss is exactly zero inside sample. Thus it must be a global minimum. $\qquad\square$

### E.8 PROOF OF THEOREM 6

*Proof.* First, we obtained new $\epsilon$-covering bound for the parameter set $(\boldsymbol{w}^{(2)}, \boldsymbol{w}^{(1)}, \boldsymbol{w}_f, \boldsymbol{w}_a)$:

$$N_{\epsilon_1} \leq (3C_1/\epsilon)^d, N_{\epsilon_2} \leq (3C_2/\epsilon)^{td}, N_{\epsilon_3} \leq (3C_8/\epsilon)^{d_a}, N_{\epsilon_4} \leq (3C_9/\epsilon)^{d_f},$$

And $N_\epsilon \leq \Pi_{i=1}^4 N_{\epsilon_i}$ Similar to Theorem 1, we denote

$$u = (\boldsymbol{w}^{(2)T}\phi(\langle\boldsymbol{w}^{(1)}, \sum_{j=1}^{p} \boldsymbol{a}(\boldsymbol{x}_i)\boldsymbol{x}_i^j\rangle) - \mathbb{E}(y|\boldsymbol{x})) \qquad (12)$$

and $u_i$ as the version with specified sample index. Then the derivatives of population risk with expection to $\boldsymbol{y}$ can be presented as follows:

$$\nabla R(\boldsymbol{w}^{(2)}) = \frac{1}{n} \sum_{i=1}^{n} u_i \phi(\langle \boldsymbol{w}^{(1)}, \boldsymbol{x}_i \odot \boldsymbol{a}\rangle)$$

$$\nabla R(\boldsymbol{w}^{(1)}) = \frac{1}{n} \sum_{i=1}^{n} u_i (\sum_{j=1}^{p} \boldsymbol{a}(\boldsymbol{x}_i)\boldsymbol{x}_i^j)(\boldsymbol{w}^{(2)} \odot \phi^{'}(\langle \boldsymbol{w}^{(1)}, \sum_{j=1}^{p} \boldsymbol{a}(\boldsymbol{x}_i)\boldsymbol{x}_i^j\rangle))^T$$

And also, the new $\boldsymbol{v}_i$ and $\boldsymbol{z}_{ik}$ terms are:

$$\boldsymbol{v}_i = \boldsymbol{u}_i \phi(\langle \boldsymbol{w}_j^{(1)}, \sum_{j=1}^{p} \boldsymbol{a}(\boldsymbol{x}_i)\boldsymbol{x}_i^j\rangle)$$

$$\boldsymbol{z}_{ik} = u_i \boldsymbol{a}(\boldsymbol{x}_{ik})\boldsymbol{x}_{ik}^j (\boldsymbol{w}^{(2)T}) \odot \phi^{'}(\langle \boldsymbol{w}_j^{(1)}, \sum_{j=1}^{p} \boldsymbol{a}(\boldsymbol{x}_i)\boldsymbol{x}_i^j\rangle))$$

In the case when $\|\mathbb{E}(\nabla R(\boldsymbol{w}^{(2)}))\|_2 \geq c\gamma$. we have new bound of $\|\boldsymbol{v}\|_2^2$ with respect to $\boldsymbol{x}$ is upper bounded by $\sigma^2 = t^2 C_1^4 C_2^2 C_8^2 C_9^2 C_X^4$ with normalized attention weight. Same argument follows for the case when $\|\mathbb{E}(\nabla R(\boldsymbol{w}^{(1)})_k\|_2 \geq c\gamma$ Then follow the same approach as Theorem 1 and 2, we obtain the sample complexity bound:

$$n \gtrsim \frac{\sigma^2}{c^2\gamma^2} \log(\frac{tC_1 C_2 C_8 C_9 C_x}{c^3\gamma})(d + td + d_f + d_a)$$

$\square$

## E.9    Proof of Theorem 7

*Proof.* Under assumption (A1) and (A14), we know all input features and weights are bounded. Therefore we know $\|\nabla f(\boldsymbol{w}^{k_r})\|_2$ is Lipschitz continuous function, and we denote its Lipschitz constant $L_k$. For $\boldsymbol{w}^{k_r}$, we can derive that:

$$\nabla R(\boldsymbol{w}^{k_r}) = \frac{1}{n} \sum_{i=1}^{n} u_i \nabla f(\boldsymbol{w}^{k_r})$$

Under (A14) to (A16), if we have $\mathbb{E}(f(\boldsymbol{x}) - \mathbb{E}(y|\boldsymbol{x}))^2 \lesssim \gamma^2$, then:

$$\|E(\nabla R(\boldsymbol{w}^{k_r}))\|_2 \gtrsim sd(f(\boldsymbol{w}^{k_r})\|_2)sd(u)cor(\nabla f(\boldsymbol{w}^{k_r}), \boldsymbol{u}) \gtrsim O(\gamma)$$

Then similar with Theorem 1 and 2, we construct an $\epsilon$-cover over all parameters $\boldsymbol{\theta} := (\boldsymbol{w}^{k_1}, \boldsymbol{w}^{k_2}, \boldsymbol{w}^V, \boldsymbol{w}^Q, \boldsymbol{w}^K)$, and we denote it as $\{\boldsymbol{\theta}_1, \ldots, \boldsymbol{\theta}_N\}$ such that for any feasible parameter, there exist $j \in [N]$ such that the maximum $\ell_2$ distance to $\boldsymbol{\theta}_j$ is smaller than $\epsilon$. By calculating the number of parameters in all matrices in $\boldsymbol{\theta}$, we have

$$\log(N_\epsilon) = \frac{1}{\epsilon}O(d_{self} + \sum_{i=1}^{k}(d_k + d_v)q_k)$$

Denoting $\nabla R(\boldsymbol{w}_{(j)}^{k_r})$ as the gradient with respect to $j^{th}$ parameter set in $\epsilon$-cover for $j \in \{1, \ldots, N_\epsilon\}$:

$$P(\|\nabla R(\boldsymbol{w}_{(j)}^{k_r}) - \mathbb{E}_x(\nabla R(\boldsymbol{w}_{(j)}^{k_r})\|_2 \geq \frac{c\gamma}{3}) \lesssim \exp(-n\frac{c^2\gamma^2}{c_k^2})$$

By union bound, we have:

$$P(\exists j \in [N_\epsilon], \|\nabla R(\boldsymbol{w}_{(j)}^{(2)})\|_2 \geq \frac{2c\gamma}{3}) \lesssim N_\epsilon \exp(-n\frac{c^2\gamma^2}{c_k^2})$$

Secondly we analyze $\|\nabla R(\boldsymbol{w}^{k_r}) - \nabla R(\boldsymbol{w}_{(j)}^{k_r})\|_2$ term. As we have shown that the gradient is Lipschitz continuous, thus we have:

$$\|\nabla R(\boldsymbol{w}^{k_r}) - \nabla R(\boldsymbol{w}_{(j)}^{k_r})\|_2 \le L_k \epsilon (d_{self} + \sum_{i=1}^{k} (d_k + d_v) q_k)$$

We choose $\epsilon = \frac{c\gamma}{3L_k}$, then at least with probability $1 - \mathcal{O}(N_\epsilon \exp(-n\frac{c^2\gamma^2}{\sigma^2}))$, we have $\|\nabla R(\boldsymbol{w}^{(2)})\|_2 > \frac{c\gamma}{3}$. Therefore we can choose $n \gtrsim \log(\frac{c_k}{\gamma})(d_{self} + \sum_{i=1}^{k}(d_k+d_v)q_k)$, such that $N_\epsilon \exp(-n\frac{c^2\gamma^2}{c_k^2}) = o(1)$. Finally we can conclude that with probability $1 - o_n(1)$, for any $(\boldsymbol{a}, \boldsymbol{w}^{(1)}, \boldsymbol{w}^{(2)})$ such that $\mathbb{E}(\tilde{\boldsymbol{w}}^{(2)}\phi(\langle \tilde{\boldsymbol{w}}^{(1)}, \boldsymbol{x} \odot \tilde{\boldsymbol{a}}\rangle) - \mathbb{E}(y|\boldsymbol{x}))^2 \ge \gamma$, we have $\|\nabla R(\boldsymbol{w}^{(2)})\|_2 > \frac{c\gamma}{3}$. Then following the convergence of empirical risk procedure of Theorem 1, we show with probability going to 1 such that $\|\nabla R_n(\boldsymbol{w}^{(2)})\|_2 > 0$ and all parameters with prediction error $O(\gamma)$ cannot be stationary point as long as $n \gtrsim \log(\frac{c_k}{\gamma})(d_{self} + \sum_{i=1}^{k}(d_k + d_v)q_k)$. Thus we complete the proof. □

## F ADDITIONAL NUMERICAL EXPERIMENTS

### F.1 CONVERGENCE PLOTS FOR CLASSIFICATION TASKS ON NOISY-MNIST DATASET

We present additional results on convergence of $2-$ layer neural networks on classification task involving Noisy-MNIST dataset (Section 5.1 of main paper). This dataset is formed by embedding an $s \times s$ digit image in a noisy image of size $48 \times 48$. We show convergence results for three settings: $s = \{5, 8, 15\}$. In each setting, the learning rate of models are varied. The plots are shown in Figures 4, 5, and 6. We observe that in every setting, attention models converge faster than the baseline models not employing attention.

### F.2 CONVERGENCE PLOTS FOR REGRESSION TASK

In this section, we present convergence plots for sample complexity experiments discussed in Section 5.3 of the main paper. Regression task is considered in this experiment. Convergence plots of baseline model, attention model and regularized attention models are plotted for various sizes of the datset $N_s$ as shown in Figure 7. We observe that for all $N_s$ values, regualrized attention model converge fastest followed by attention model which is then followed by baseline model. So, we conclude that in addition to achieving improved sample complexity, attention models converge faster than models not employing attention. Also, regularization helps speed up the model convergence.

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

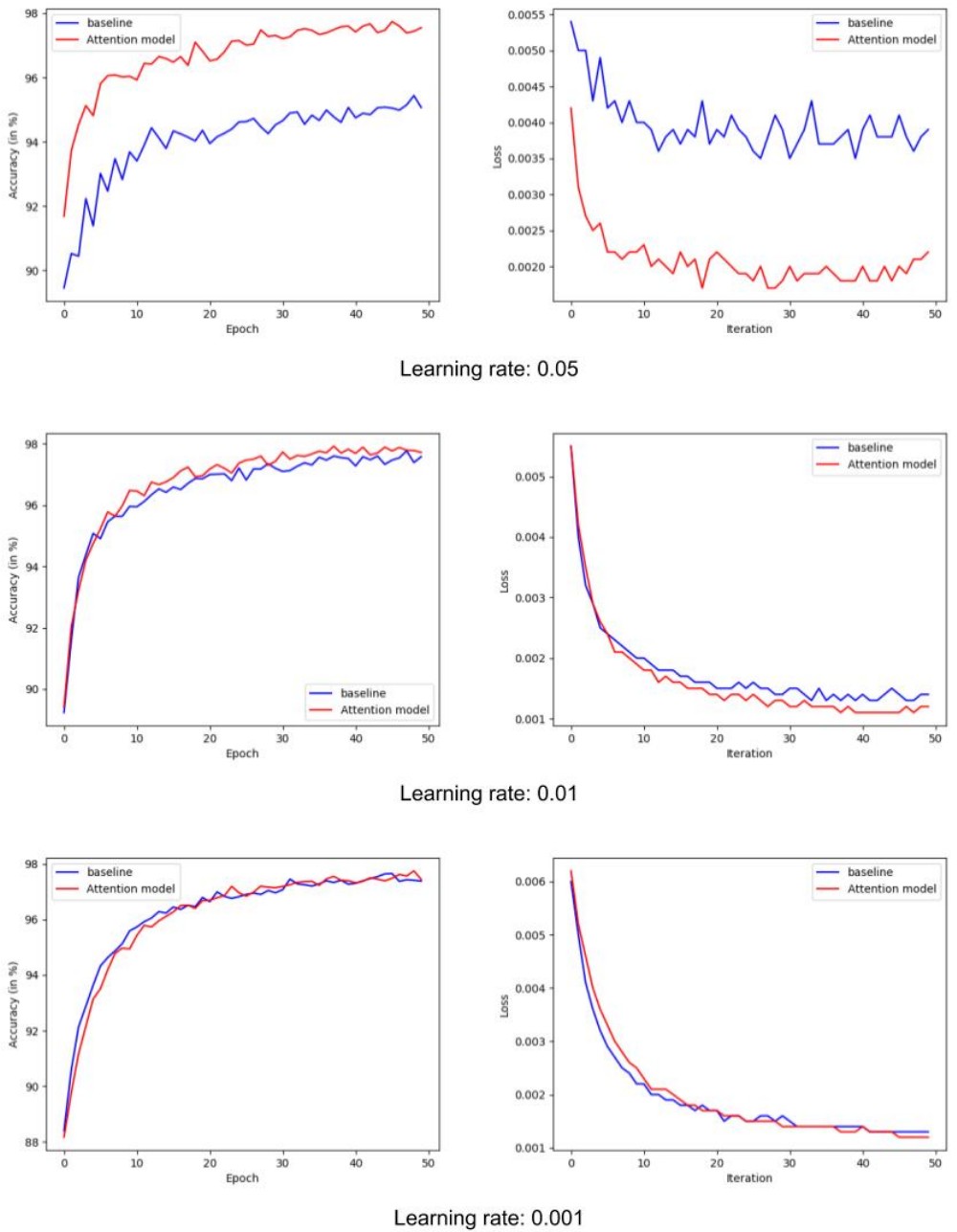

Figure 4: Convergence plots for classification tasks: Image size: $48 \times 48$, Digit patch size $s = 15$

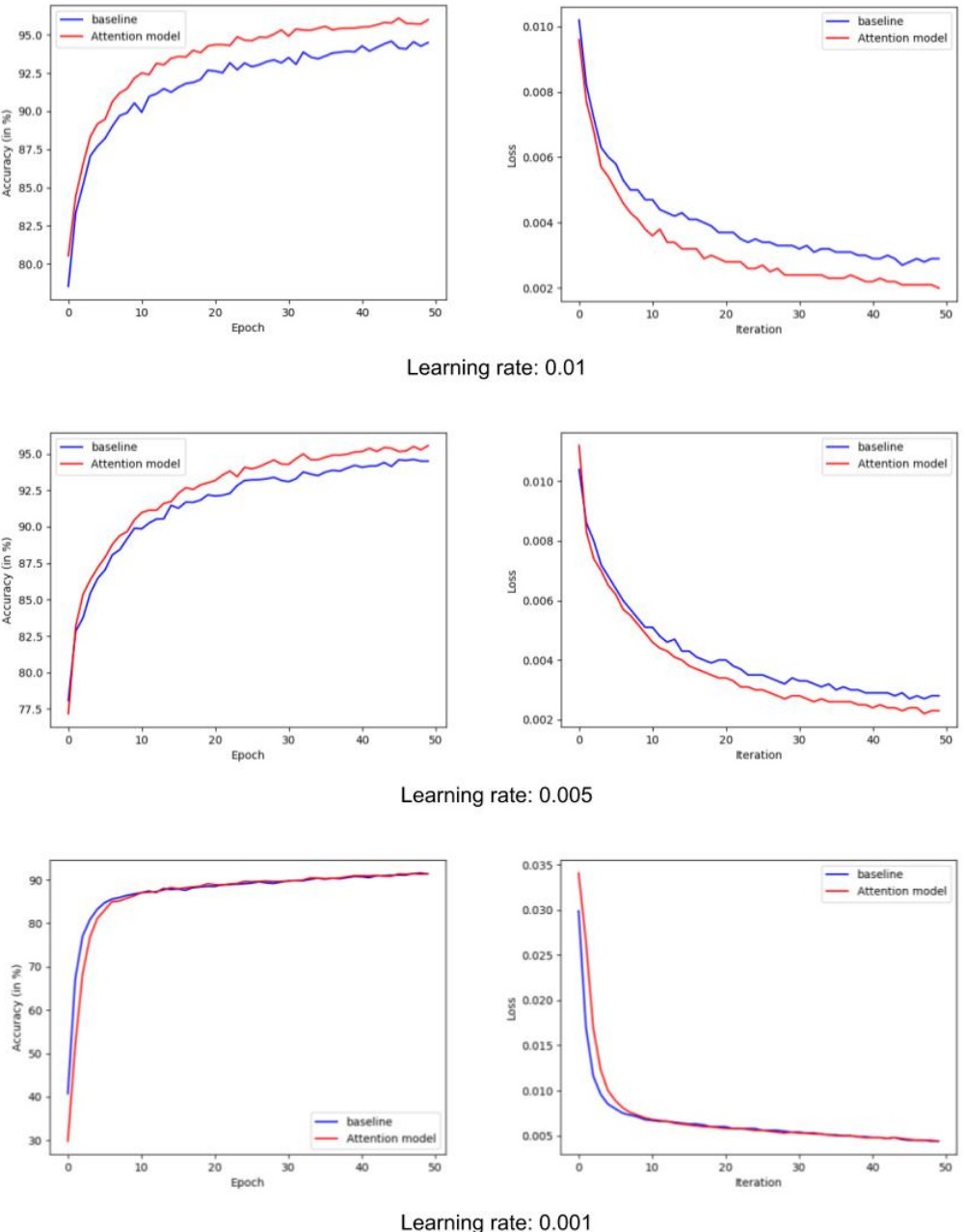

Figure 5: Convergence plots for classification tasks: Image size: $48 \times 48$, Digit patch size $s = 8$

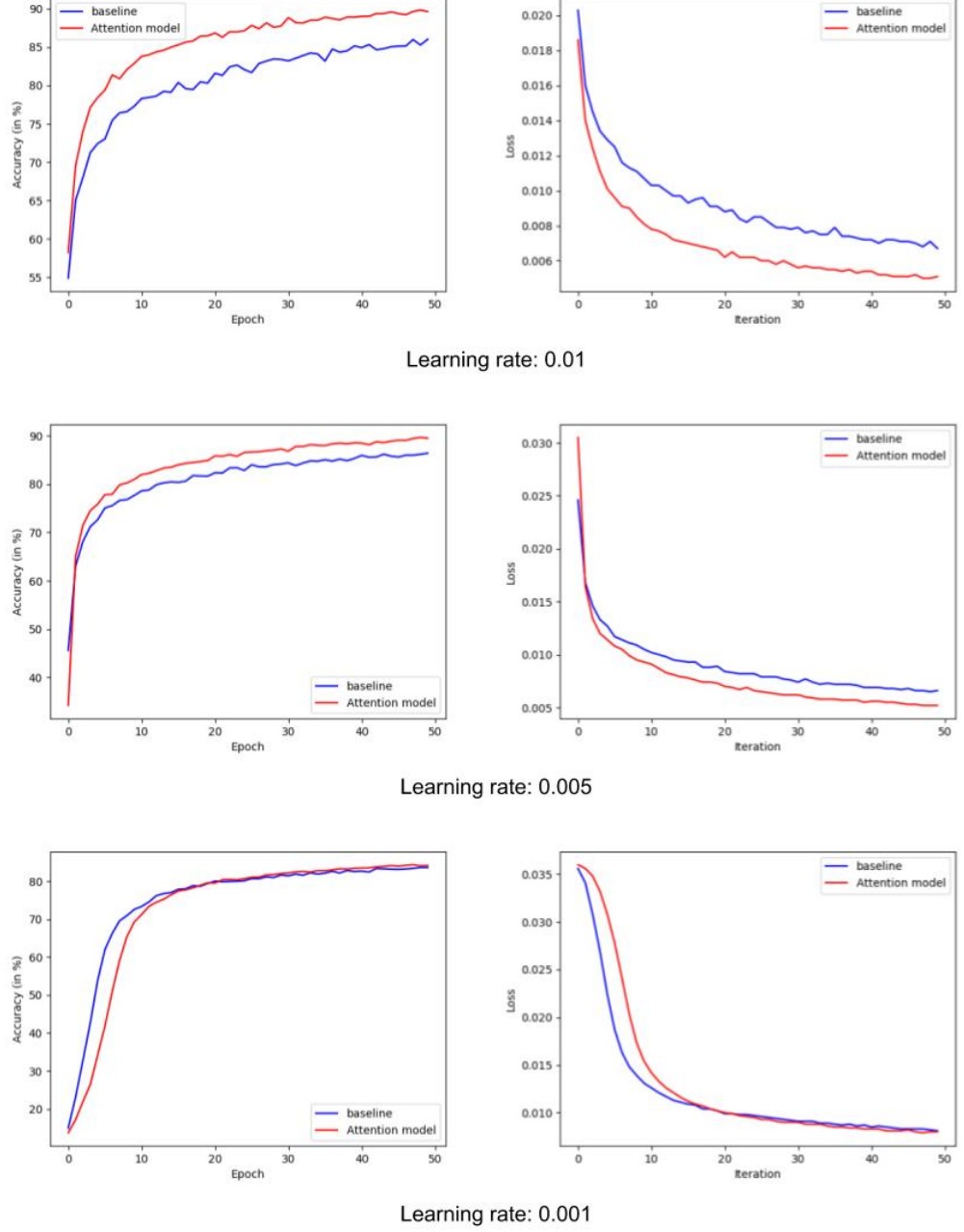

Figure 6: Convergence plots for classification tasks: Image size: $48 \times 48$, Digit patch size $s = 5$

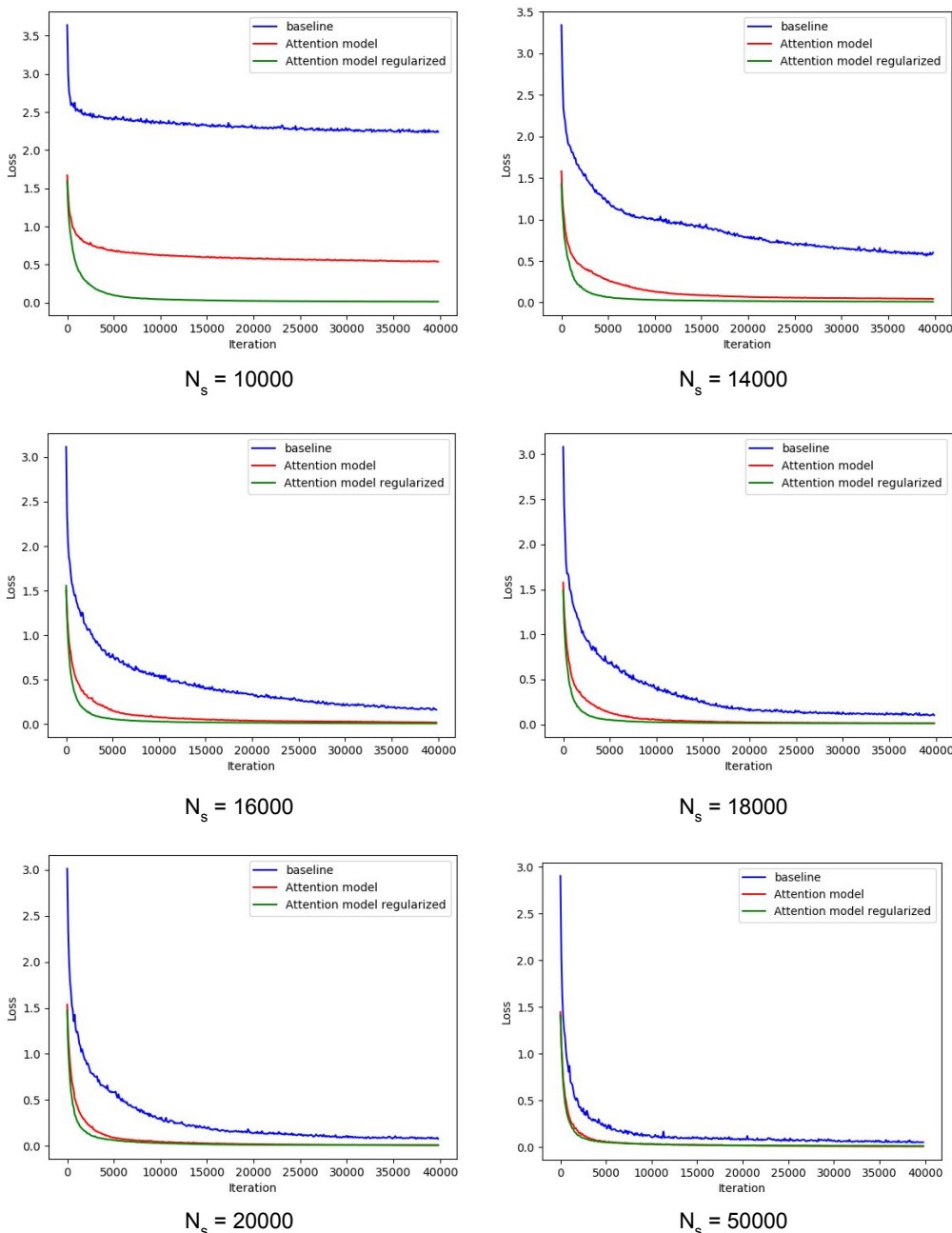

Figure 7: Convergence plots for regression task varying the number of samples $N_s$. "Baseline" indicates $2-$ layer NN not using attention, "attention" denotes attention models, and "attention regualrized" denotes attention models trained with $L_1$ regularization

Simon S Du, Jason D Lee, Haochuan Li, Liwei Wang, and Xiyu Zhai. Gradient descent finds global minima of deep neural networks. In *Proceedings of the 36th International Conference on Machine Learning*, ICML '19, 2019.

Rong Ge, Jason D. Lee, and Tengyu Ma. Learning one-hidden-layer neural networks with landscape design. *CoRR*, abs/1711.00501, 2017.

Xavier Glorot, Antoine Bordes, and Yoshua Bengio. Deep sparse rectifier neural networks. In *Proceedings of the fourteenth international conference on artificial intelligence and statistics*, pp. 315–323, 2011.

Ian Goodfellow, Yoshua Bengio, and Aaron Courville. *Deep learning*. MIT press, 2016.

Sepp Hochreiter and Jürgen Schmidhuber. Flat minima. *Neural Computation*, 9(1):1–42, 1997.

Nitish Shirish Keskar, Dheevatsa Mudigere, Jorge Nocedal, Mikhail Smelyanskiy, and Ping Tak Peter Tang. On large-batch training for deep learning: Generalization gap and sharp minima. *arXiv preprint arXiv:1609.04836*, 2016.

Jiasen Lu, Jianwei Yang, Dhruv Batra, and Devi Parikh. Hierarchical question-image co-attention for visual question answering. In *NIPS*, pp. 289–297, 2016.

Thang Luong, Hieu Pham, and Christopher D. Manning. Effective approaches to attention-based neural machine translation. In *Proceedings of the 2015 Conference on Empirical Methods in Natural Language Processing*, pp. 1412–1421. Association for Computational Linguistics, 2015.

Andrew L. Maas, Raymond E. Daly, Peter T. Pham, Dan Huang, Andrew Y. Ng, and Christopher Potts. Learning word vectors for sentiment analysis. In *Proceedings of the 49th Annual Meeting of the Association for Computational Linguistics: Human Language Technologies*, pp. 142–150, Portland, Oregon, USA, June 2011. Association for Computational Linguistics. URL http://www.aclweb.org/anthology/P11-1015.

Song Mei, Yu Bai, Andrea Montanari, et al. The landscape of empirical risk for nonconvex losses. *The Annals of Statistics*, 46(6A):2747–2774, 2018a.

Song Mei, Andrea Montanari, and Phan-Minh Nguyen. A mean field view of the landscape of two-layer neural networks. *Proceedings of the National Academy of Sciences*, 115(33):E7665–E7671, 2018b. ISSN 0027-8424. doi: 10.1073/pnas.1806579115.

Volodymyr Mnih, Nicolas Heess, Alex Graves, and koray kavukcuoglu. Recurrent models of visual attention. In Z. Ghahramani, M. Welling, C. Cortes, N. D. Lawrence, and K. Q. Weinberger (eds.), *Advances in Neural Information Processing Systems 27*, pp. 2204–2212. Curran Associates, Inc., 2014.

Guido F Montufar, Razvan Pascanu, Kyunghyun Cho, and Yoshua Bengio. On the number of linear regions of deep neural networks. In *Advances in neural information processing systems*, pp. 2924–2932, 2014.

Quynh Nguyen and Matthias Hein. The loss surface of deep and wide neural networks. In *Proceedings of the 34th International Conference on Machine Learning (ICML 2017), Sydney, NSW, Australia, 6-11 August 2017*, pp. 2603–2612, 2017a.

Quynh Nguyen and Matthias Hein. The loss surface of deep and wide neural networks. *arXiv preprint arXiv:1704.08045*, 2017b.

Tomaso A. Poggio and Qianli Liao. Theory II: landscape of the empirical risk in deep learning. *CoRR*, abs/1703.09833, 2017.

Yunchen Pu, Martin Renqiang Min, Zhe Gan, and Lawrence Carin. Adaptive feature abstraction for translating video to text. In *Proceedings of the Thirty-Second AAAI Conference on Artificial Intelligence, (AAAI-18), New Orleans, Louisiana, USA, February 2-7, 2018*, pp. 7284–7291, 2018.

Blaine Rister and Daniel L. Rubin. Piecewise convexity of artificial neural networks. *Neural Networks*, 94:34–45, 2017.

Mahdi Soltanolkotabi, Adel Javanmard, and Jason D. Lee. Theoretical insights into the optimization landscape of over-parameterized shallow neural networks. *CoRR*, abs/1707.04926, 2017.

Daniel Soudry and Yair Carmon. No bad local minima: Data independent training error guarantees for multilayer neural networks. *arXiv preprint arXiv:1605.08361*, 2016.

Daniel Soudry and Elad Hoffer. Exponentially vanishing sub-optimal local minima in multilayer neural networks, 2018.

Ashish Vaswani, Noam Shazeer, Niki Parmar, Jakob Uszkoreit, Llion Jones, Aidan N Gomez, Łukasz Kaiser, and Illia Polosukhin. Attention is all you need. In *Advances in Neural Information Processing Systems*, pp. 5998–6008, 2017.

Roman Vershynin. Introduction to the non-asymptotic analysis of random matrices. *arXiv preprint arXiv:1011.3027*, 2010.

Kelvin Xu, Jimmy Ba, Ryan Kiros, Kyunghyun Cho, Aaron Courville, Ruslan Salakhudinov, Rich Zemel, and Yoshua Bengio. Show, attend and tell: Neural image caption generation with visual attention. In Francis Bach and David Blei (eds.), *Proceedings of the 32nd International Conference on Machine Learning*, volume 37 of *Proceedings of Machine Learning Research*, pp. 2048–2057, Lille, France, 07–09 Jul 2015. PMLR.

Han Zhang, Ian J. Goodfellow, Dimitris N. Metaxas, and Augustus Odena. Self-attention generative adversarial networks. *CoRR*, abs/1805.08318, 2018.

Bolei Zhou, Yuandong Tian, Sainbayar Sukhbaatar, Arthur Szlam, and Rob Fergus. Simple baseline for visual question answering. *arXiv preprint arXiv:1512.02167*, 2015.

Pan Zhou and Jiashi Feng. The landscape of deep learning algorithms. *CoRR*, abs/1705.07038, 2017.

