# OpenReview forum: "Understanding Attention Mechanisms"
_ICLR.cc/2020/Conference — Reject_

### Official Review · AnonReviewer3 · 2019-10-23
**Official Blind Review #3**

**Rating:** 6

**Review:**

This paper studies attention and self-attention networks from the theoretical perspective, giving the first (as far as this reviewer knows) results proving that attention networks can generalize better than non-attention baselines. This has been observed empirically before and it is very good to start the analysis of foundations of this phenomenon.

The results cover fixed-attention (as we understand both single-layer and multi-layer) and self-attention in the single layer setting. One interesting part that is not covered (and may be very hard) though is multi-layer self-attention networks. What is the equivalent of condition (A9) there? In other words: can we prove that the attention will learn correct attention masks if they exist, or do we need to assume that?

On the experimental side, the paper introduces L1 loss on the attention mask. This is (to the knowledge of this reviewer) a new idea and it would be interesting to see larger models (e.g., a Transformer on a translation task) trained with this additional loss. Does the analysis suggest L1 loss in any way, more than say L2?

I am grateful to the authors for their reply. The new multi-layer theorem is impressive and I'm grateful for clarifying the L1 loss. Have the authors considered variants of hard attention as well (e.g., training with attention that attends to at most 10 or 100 elements), could that play the role of L1 loss? I stand by my recommendation to accept this paper.

**Experience Assessment:**

I have published one or two papers in this area.

**Review Assessment: Checking Correctness Of Derivations And Theory:**

I assessed the sensibility of the derivations and theory.

**Review Assessment: Checking Correctness Of Experiments:**

I carefully checked the experiments.

**Review Assessment: Thoroughness In Paper Reading:**

I read the paper thoroughly.

---

> ### Author Response · Authors · 2019-11-13
> **Detailed responses to reviewer 3**
>
> Thank you for your helpful suggestions. In what follows, we repeat your  comments and explain how we addressed your questions.
>
> Question 1: 'One interesting part that is not covered (and may be very hard) though is multi-layer self-attention networks. What is the equivalent of condition (A9) there? In other words: can we prove that the attention will learn correct attention masks if they exist, or do we need to assume that?'
>
> Answer: Thanks for suggesting the analysis of multi-layer self-attention networks. In the updated version, we provide a more rigorous theorem of sample complexity bound for multi-layer self-attention network. Please see Theorem 7 in Appendix D.2 for more details. The new Theorem 7 indicates that we can learn correct attention masks if they exist, under assumption parallel to (A8) to (A10), without additional assumptions.
>
> As the remark of theorem 7 says, because multi-layer self-attention models include a large parameter set with complicated gradients, the assumptions are not as intuitive as the two-layer model. But the main assumptions are parallel, such as the bias $u_i$ not being uncorrelated with all possible directions. This assumption is mild, considering the high-dimensional nature of networks. The extension of this multi-layer model is omitted in the main paper since it leads to complicated derivations and complicated assumption discussion and will distract readers from the main ideas of the paper. We believe the two-layer attention model is representative enough to provide theoretical evidence on why attention reduces sample complexity.
>
> Question 2: 'On the experimental side, the paper introduces L1 loss on the attention mask. Does the analysis suggest L1 loss in any way, more than say L2?'
>
> Answer: Thanks for pointing out that we should discuss more on the choice of regularization on attention. The main reason that L1 loss is more helpful is that it can reduce the sparsity level of attention masks. We have added the following paragraph of discussion in Appendix D.1.(In page 15-16)
>
> ‘Through our theorem 1, we can tell why L1 loss is more helpful than L2 loss under sparsity assumption. Both L1 loss and L2 loss will control the magnitude of attention masks. However, L1 loss can also help control sparsity level $s_0$, and we can see that the sample complexity bound is proportional to $s_0^2$. Although L2 loss on attention mask can also help reduce the sample complexity, its effect will only be reflected in the $\log$ term, therefore it is less effective. Similarly, in Theorem 2, if we assume the attention to be sparse(i.e. one word should not attend to all the words, but only some relevant words), then L1 loss can help further reduce the sample complexity.’

---

### Official Review · AnonReviewer1 · 2019-10-25
**Official Blind Review #1**

**Rating:** 3

**Review:**

Summary: This paper endeavors to develop some theoretical understanding of why attention mechanisms improve the performance of neural networks. It includes some theory around the sample complexity of models which have a multiplicative mask in them, as well as a few related experiments.

Review: Having some theoretical analysis of the benefits of attention is definitely needed and I commend the authors for working towards this goal. The main issue that I have with the paper is that the model initially considered (introduced in section 2) is unrelated to attention mechanisms because the attention weights "a" do not depend on the input in any way. If I understand correctly, the ground-truth function y_i = f*(a* . x_i) assumes there is a single globally-correct attention mask a* for all examples. This is not attention. Every attention mechanism that I am aware of depends on the input in some way (in other words, instead of a*, you'd have a function a*(x_i) which changes according to the input). Some additional discussion is needed to explain why it is interesting to study this dramatically simplified (to the point of not being relevant) model. For example, if it was explicitly described as a preliminary for the a*(x_i) case (section 3.2 onwards) it would make more sense. Similarly, the assumption ||a_0|| ≤ s_0 is not true for most attention masks in pratice. There are some hard attention mechanisms which try to induce binary masks/hard sparsity but they are extremely rarely used in practice. The combination of these two assumptions makes for what appears to me (if I'm understanding correctly) a somewhat trivial result that the "attention mechanisms constrain the parameters in a smaller space". Note that this criticism does not apply to section 3.2 where the attention mask is specified as a function of the input. Similarly section 4.2 relies on the "single attention mask" idea, which is the sole result considering generalization in the paper. Finally 5.1.1 consists of an experiment which assumes a model of the form G*(x . a*), where a* is a fixed random binary mask. Training a model to learn G* and a* has very little to do with attention - it is a learning problem where you have removed a subset of the features and are trying to determine the parameters of the function and which features were removed. The L_1 penalty on "a" is certainly a good way to learn to only include a sparse subset of your features, but again has very little relevance to most attention mechanisms (i.e. I know of almost no work which regularizes the attention mask in this way).

Overall I think most of the results in this paper are not relevant to attention models and as such it has limited impact. I would suggest reframing the paper around a focus on what the authors call the "self-attention" case (which is really just normal attention), possibly using the fixed a* case as a preliminary/motivation. I also would suggest removing the discussion of regularization the attention mask as I am not aware of this being used in practice. Finally I would suggest some more direct discussion of how the results apply to models which are used in practice.

**Experience Assessment:**

I have published in this field for several years.

**Review Assessment: Checking Correctness Of Derivations And Theory:**

I did not assess the derivations or theory.

**Review Assessment: Checking Correctness Of Experiments:**

I assessed the sensibility of the experiments.

**Review Assessment: Thoroughness In Paper Reading:**

I read the paper thoroughly.

---

> ### Author Response · Authors · 2019-11-13
> **Detailed responses to reviewer 1**
>
> Thank you for your helpful suggestions and comments. In general, we agree that the naive global attention model is not very practical, but we argue that it is very relevant to general attention models. In the revised draft, we renamed the "global attention" as "naive global attention" to make it clear for the readers that it does not really capture the essence of attention mechanisms. And we also explicitly state the importance of analyzing this model, as it is a starting point for analyzing self-attention models. Please see the 'summary of major changes in the revised version'  we made in the revised paper.
>
> In what follows, we repeat your comments and explain how we addressed them.
>
> Question 1: 'The model initially considered (introduced in section 2) is unrelated to attention mechanisms because the attention weights "a" do not depend on the input in any way.'
>
> Answer: We agree the global attention model we analyze in Section 3.1 is not very practical in real-world applications. However, it is an important starting point for analyzing attention mechanisms for two reasons. We have added the following two paragraphs in the new Section 2 (In page 2-3):
>
> ‘First, this fixed attention shares the core idea of attention: There is a specific intrinsic structure in data, and this intrinsic structure requires that we should assign different weights to different input features accordingly. And this weight assigning strategy should be learned from data. In naive global attention, attention weights ${a}$ are parameters themselves; In the self-attention model, the attention weights ${a}$ depend on a function parameterized by value/key/query matrices, and we learn these matrices as attention parameters. And for both global and self-attention, we jointly learn the network weights(${w}^{(1)}$,${w}^{(2)}$) and attention parameters($a$ in global attention, value/key/query matrices in self-attention) at the same time. Therefore this naive global attention is a good starting point for analyzing attention mechanisms.’
>
> ‘Second, we can gain helpful insights on attention by analyzing the global attention case. The number of non-zero elements of $a$ in global attention represents both the size of attention parameters and the sparsity level of attention. In the standard self-attention model, the size of attention parameters is determined by the size of value/key/query matrices. And the sparsity level is how many words we allow one word to attends to. By studying the effect of this quantity, we can have a better understanding of how the sparsity and parameter size of attention affect the model performance and sample complexity. The detailed discussions can be found in Section 3.’
>
> Question 2: 'The assumption $\|a_0\| \leq s_0$ is not true for most attention masks in practice. There are some hard attention mechanisms which try to induce binary masks/hard sparsity but they are extremely rarely used in practice. '
>
> Answer: We think analyzing the sparsity of attention is still important. Firstly, although hard attention is less popular than soft attention because of its non-differentiable nature, it is still widely applied because it has a concrete interpretation and is desirable in many applications such as Xu et al. (2015), Luong et al. (2015), and Pu et al. (2018). And these hard attention models enjoy the sparsity of attention; Secondly, please see the second reason under Question 1, it implies why sparsity helps understand the sample complexity of attention models.
>
> Question 3: 'Similarly section 4.2 relies on the "single attention mask" idea, which is the sole result considering generalization in the paper.'
>
> Answer: We have extended Theorem 4 in section 4.2 and Theorem 5 in section 4.3 to self-attention models. Please check the new theorems in our revised paper. Also, our major sample complexity bound results in Section 3 also consider generalization of models with and without attention.
>
>
> Question 4: 'Finally 5.1.1 consists of an experiment which assumes a model of the form G*(x $\odot$ a*), where a* is a fixed random binary mask.'
>
> Answer: We would like to point out that empirical evaluation in Section 5.1.1 is not a result in itself. It was performed to supplement our theoretical findings. Hence, the assumptions considered here mimics the assumptions made in the theorem. Section 5.1.1 and Section 5.1.2 validates the theoretical claims about global attention and self-attention empirically.
>
>
> Question 5: 'Finally I would suggest some more direct discussion of how the results apply to models which are used in practice.'
>
> Answer: In this paper, we consider self-attention and recurrent attention structures, according to Vaswani et al. (2017) and Bahdanau et al. (2014) separately. And we also verify the self-attention results empirically on the IMDB dataset.

---

> > ### Comment · AnonReviewer1 · 2019-11-15
> > **Response**
> >
> > Thank you for your clarifications. I think you have improved the positioning of your work but I remain convinced that the following statement is not true:
> > >  naive global attention is a good starting point for analyzing attention mechanisms.
> > What you call naive global attention has very little to do with attention. I emphasize my suggestion that you remove discussion of "naive global attention" from your paper and focus on the case that you currently call "self-attention". This will make a much stronger submission.
> >
> > Specifically to your point about sparseness,
> > > it is still widely applied because it has a concrete interpretation and is desirable in many applications such as Xu et al. (2015)
> > One of the takeaways from Xu et al. is that the hard attention is actually significantly less interpretable than the soft attention; the hard attention mechanism often attends to parts of the image which are irrelevant to the current word.

---

### Official Review · AnonReviewer2 · 2019-10-26
**Official Blind Review #2**

**Rating:** 3

**Review:**

This paper studies the loss landscape of two-layer neural networks on global- and self-attention models.  It shows that attention helps reduce sample complexity.  I went through the all the theorem part of the paper, but only checked the intuition and did not dig into the detailed proof.   I am less familiar with reviewing papers that are theorem-oriented, which is hard for me to justify the contributions of these bounds.  My concerns and suggestions are mainly focused on the empirical part, which I think authors need to improve.

First, the authors need to provide more detail about the baseline model, especially for the experiment on 5.1.1.  Also, please list the number of parameters for different models.  And please justify whether the reported results (sample complexity in table 1 & 2) are affected by the difference of number parameters if they are not the same.  The authors also need to clarify the meaning of the "number of training samples" in the tables.  Does it mean the number of unique training examples seen by the model or the total number of training samples?  If the authors mean the former one, then, what is the training procedure?  What are optimizers used between different models?  Are these will affect the test loss difference.  How does the author make sure the comparisons are fair?  Also, in the main text of section 5.1.1, the authors said they construct a dataset of 200,000 examples.  How are the 200,000 samples distributed?  What is the size of the testing set?   Do the reported results average over multiple runs? Similarly, in section 5.1.2 authors need to clarify these questions and the experiments are fair to support their claim.

Given the theoretical part is a contribution, the experiments at the current version do not support their claim fully.  I will consider raising my scores if authors can clarify my questions.

**Experience Assessment:**

I have read many papers in this area.

**Review Assessment: Checking Correctness Of Derivations And Theory:**

I did not assess the derivations or theory.

**Review Assessment: Checking Correctness Of Experiments:**

I assessed the sensibility of the experiments.

**Review Assessment: Thoroughness In Paper Reading:**

I made a quick assessment of this paper.

---

> ### Author Response · Authors · 2019-11-13
> **Detailed responses to reviewer 2**
>
> Thank you for your valuable comments. More details regarding the experiments are included in Section 5 of the revised draft. For both global attention and self-attention experiments, the number of parameters used are included in parenthesis. We note that the number of parameters of attention models and baseline models are comparable. Additionally, in Table 2, we also performed an experiment where we used 2x the number of parameters for baseline model as compared to the self-attention model. We observe that even with 2x parameter size, baseline models perform poorly compared to self-attention. This shows that sample complexity gains obtained are a consequence of attention mechanisms.
>
> Number of training samples: We apologize for this confusion. Please ignore the statement on 200k samples as was mentioned in the previous draft. In Section 5.1.1, we updated the description of dataset creation to make it more clear. To test sample complexity, we generate multiple datasets, each containing 10k, 14k, 16k, 18k, 20k and 50k unique labeled samples respectively, generated according to the procedure mentioned in Sec. 5.1.1. Then, each model is trained only on the specific dataset. For instance, a model trained on 10k dataset only sees 10k unique samples. It is however trained for many iterations, hence it loops over the same 10k samples after each epoch is complete. All models are tested on a common test set of size 5000 samples, which is also generated using the same procedure as described in Sec. 5.1.1.
>
> For global attention experiments, SGD optimizer with learning rate 0.001 is used. For self-attention experiments, adam optimizer with learning rate 0.001 is used. These configurations gave the best performance among other learning rate settings we tried. However, we would like to point out that the observation that attention models need a lower sample complexity than baseline models hold for all learning rate configurations. We provided details of the optimizers in the updated draft.
>
> We would like to emphasise that all assumptions made are in-line with the assumptions made in the Theory section. Hence, these results supplement our theoretical findings.

---

### Author Response · Authors · 2019-11-13
**Summary of major changes in revised version**

We thank all the reviewers for the constructive comments and valuable suggestions. We have uploaded a revised version of our paper following the suggestions. In the revised paper, we have made the following changes in each section.

In Section 2, we introduce both naive global attention and self-attention model setup in the new version. Here we specify that naive global attention has a fixed attention mask, and is not practically used. Then we state reasons why it should be treated as a simple attention model and very important to be analyzed first. Analysis of naive global-attention models is a starting point, and helps analyze self-attention models.

In Section 3, we further discuss the connection between naive global attention and self-attention in the "Remark" section under Theorem 2.

In Section 4, we extend the result of Theorem 4 (on flatness of minima) and Theorem 5 (on small sample size) to self-attention models.

In Section 5, we add more experiment details about the number of trainable parameters, optimization hyper-parameters and other details validating the fairness of our experiments.

In Appendix D.1., we add more detailed discussions on why L1 regularization on attention weights is more helpful than L2 regularization.

In Appendix D.2., we add a new theorem for sample complexity bound of multi-layer self-attention models.

We mark all the changes in red so that the reviewers can easily identify them in the paper. Thanks again for the invaluable time and suggestions of all reviewers. We hope you find that the revised manuscript is improved.

---

### Decision · Program_Chairs · 2019-12-19

**Decision:**

Reject

**Comment:**

This paper aims to theoretically understand the the benefit of attention mechanisms. The reviewers agreed that better understanding of attention mechanisms is an important direction. However, the paper studies a weaker form of attention which does not correspond well to the attention models using in the literature. The paper should better motivate why the theoretical results for this restrained model would carry over to more realistic mechanisms.